# A plant cell death-inducing protein from litchi interacts with *Peronophythora litchii* pectate lyase and enhances plant resistance

Wen Li [1,4], Peng Li [1,4], Yizhen Deng[2], Junjian Situ [1], Zhuoyuan He [3], Wenzhe Zhou[1], Minhui Li[1], Pinggen Xi[1], Xiangxiu Liang[3], Guanghui Kong [1]✉ & Zide Jiang [1]✉

Cell wall degrading enzymes, including pectate lyases (PeLs), released by plant pathogens, break down protective barriers and/or activate host immunity. The direct interactions between PeLs and plant immune-related proteins remain unclear. We identify two PeLs, PlPeL1 and PlPeL1-like, critical for full virulence of *Peronophythora litchii* on litchi (*Litchi chinensis*). These proteins enhance plant susceptibility to oomycete pathogens in a PeL enzymatic activity-dependent manner. However, LcPIP1, a plant immune regulator secreted by litchi, binds to PlPeL1/PlPeL1-like, and attenuates PlPeL1/PlPeL1-like induced plant susceptibility to *Phytophthora capsici*. LcPIP1 also induces cell death and various immune responses in *Nicotiana benthamiana*. Conserved in plants, LcPIP1 homologs bear a conserved "VDMASG" motif and exhibit immunity-inducing activity. Furthermore, SERK3 interacts with LcPIP1 and is required for LcPIP1-induced cell death. NbPIP1 participates in immune responses triggered by the PAMP protein INF1. In summary, our study reveals the dual roles of PlPeL1/PlPeL1-like in plant-pathogen interactions: enhancing pathogen virulence through PeL enzymatic activity while also being targeted by LcPIP1, thus enhancing plant immunity.

Plant cell walls are one of the major barriers that protect host against invading pathogens[1]. Accordingly, plant pathogens including bacteria, oomycetes, fungi, and nematodes secrete a variety of cell wall-degrading enzymes (CWDEs) in the process of infecting host plants[2,3]. CWDEs, as virulence factors, contribute to pathogenesis by destroying wax, cuticle, and cell walls, and by obtaining nutrients for supporting microbial growth, reproduction, and proliferation in the host[4]. Previous studies have revealed that silencing or deleting CWDE-encoding genes, e.g., endoxylanase genes from *Magnaporthe oryzae*, pectin methylesterase gene from *Botrytis cinerea* (*Bcpme1*), poly-galacturonase genes from *Xanthomonas campestris* (*pghAxc* and

*pghBxc*), or polygalacturonase gene from *Phytophthora capsici* (*pcipg2*), significantly reduced microbial pathogenicity[5–8]. However, plant cells can sense the alteration of cell wall integrity during biotic stress, and recognize CWDEs as pathogen-associated molecular patterns (PAMPs) or sense fragments of plant cell walls such as those degraded by CWDEs as danger-associated molecular patterns (DAMPs)[9]. Plant defense responses are thus triggered including transcriptional reprogramming, production of reactive oxygen species (ROS), cell wall fortification, and programmed cell death (PCD)[10].

Pectin is one of the major components of plant cell walls and is a structurally and functionally complex polysaccharide[11]. Pectate lyase

[1]National Key Laboratory of Green Pesticide/Guangdong Province Key Laboratory of Microbial Signals and Disease Control, South China Agricultural University, Guangzhou, China. [2]State Key Laboratory for Conservation and Utilization of Subtropical Agro-Bioresources/Integrative Microbiology Research Center, South China Agricultural University, Guangzhou, China. [3]State Key Laboratory for Conservation and Utilization of Subtropical Agro-Bioresources/ College of Life Sciences, South China Agricultural University, Guangzhou, China. [4]These authors contributed equally: Wen Li, Peng Li. ✉e-mail: gkong@scau.edu.cn; zdjiang@scau.edu.cn

(PeL) (EC 4.2.2.2) is widely distributed in plant pathogens, and is a type of pectinase. PeLs cleave β(1–4) links between galacturonosyl residues by β-elimination, resulting in 4,5-unsaturated double bonded oligogalacturonates[12]. This enzymatic action results in the rapid disintegration of pectin, leading to irreparable damage to plant tissues. The fungal PeL gene was firstly cloned from *Aspergillus nidulans*[13], after which the functions of PeLs from many plant pathogens were characterized. For example, the PeL gene *pecCl1* is an important factor in the aggressiveness of *Colletotrichum lindemuthianum*[14]. *VdPEL1* of *Verticillium dahliae* induces cell death and plant resistance, and *VdPEL1* deletion strains are severely compromised in virulence, indicating that *VdPEL1* is also a virulence factor[15]. Similarly, PcPL1, PcPL16, and PcPL20 induce cell death and contribute to the virulence of *Ph. capsici*[16]. PeLs act as virulence factors in pathogens and also induce host defense responses[17]. However, the molecular mechanisms of fungal PeL-induced plant cell death or resistance are still largely unknown.

Litchi (*Litchi chinensis* Sonn.) is a tropical and subtropical fruit with appealing color, distinct sweet taste, and abundant nutrients[18]. Downy blight caused by *Peronophythora litchii* is one of the most destructive diseases of litchi, causing enormous commercial losses[19]. Numerous CWDE genes have been reported in species of *Phytophthora* and *Pythium*[2,20], and the number of CWDEs in *P. litchii* is lower than those in some *Phytophthora* species[21]. At present, only the function of pectin acetylesterase PAE5, which is associated with the virulence of *P. litchii*, has been reported[22], and an understanding of the function of PeLs in *P. litchii* is still lacking.

Programmed cell death (PCD) plays a key role in development, cellular homeostasis, and immunity of plants[23]. In plant-microbe interactions, pathogens may trigger rapid, localized plant cell death at the site of primary infection, which is known as hypersensitive response (HR). HR blocks the nutrient absorption and spread of pathogens[24]. The reactive oxygen species (ROS) burst usually erupts at the early stage of defense response and is usually important for initiation of HR with accumulation of salicylic acid (SA), jasmonic acid (JA), and ethylene (ET)[25,26]. In previous studies, many plant components involved in PCD and resistance in *N. benthamiana* leaves have been identified, such as the class-II ethylene-responsive element binding factor NbCD1, a small orphan protein Xa7, mitogen activated protein kinase kinase NbMKK1, hypersensitive induced reaction proteins HIRs, hypersensitive response-like lesion inducing protein 4 (NbHRLI4), and a variety of transcription factors[27–33]. SERK3, which was once named BRASSINOSTEROID INSENSITIVE 1-associated receptor kinase 1 (BAK1), is an essential co-receptor of multiple receptor complexes. SERK3 is indispensable for regulation of defense-related PCD[34–36].

In this study, we investigated the biological roles of the PeL gene family in *P. litchii* and found that PlPeL1 and PlPeL1-like were required for the full pathogenicity of *P. litchii*. Furthermore, a litchi cell death-inducing protein, LcPIP1 was identified to interact with PlPeL1/PlPeL1. LcPIP1 induced plant immune responses in a SERK3-dependent manner, and attenuated the PlPeL1/PlPeL1-like- induced plant susceptibility to *Ph. capsici*. These results provide insights into plant-oomycete interactions mediated by pathogen PeLs and the mechanism of plant immunity.

## Results

### PlPeL1 and PlPeL1-like contribute to *P. litchii* virulence

We identified 19 PeL encoding genes in the *P. litchii* genome based on bioinformatic analysis. We were particularly interested in a pair of PeLs arranged in a head-to-head reversed complement fashion in the genome (Supplementary Fig. 1a), which shared high (94%) amino acid identity (Supplementary Fig. 1b). We named these two PeL encoding genes as *PlPeL1* and *PlPeL1-like*. Both PlPeL1 and PlPeL1-like have a secretion signal peptide (SP) and an Amb_all domain (SM00656) (Supplementary Fig. 1b). The function of the SP of PlPeL1/PlPeL1-like was evaluated using yeast secretion assay[37] (Fig. 1a). In addition, PlPeL1

and PlPeL1-like were expressed in *N. benthamiana* and could be detected in the plant apoplastic fluid (AF) (Fig. 1b), indicating that PlPeL1 and PlPeL1-like were secreted proteins. We next analyzed the expression patterns of *PlPeL1/PlPeL1-like* genes by qRT-PCR, and the results showed that *PlPeL1/PlPeL1-like* were up-regulated at infection stages (Supplementary Fig. 1c, d). In addition, we conducted a comparative analysis of the expression levels of *PlPeL1* and *PlPeL1-like* (Supplementary Fig. 1e).

To assess the potential contribution of *PlPeL1* and *PlPeL1-like* to *P. litchii* virulence, CRISPR/Cas9-mediated genome editing technology[38,39] was used to create individual knockout strains for *PlPeL1* and *PlPeL1-like* in *P. litchii* wild-type (WT) strain SHS3 (Supplementary Fig. 2a, b). We also generated Δ Δ *plpel1/plpel1-like* (*PlPeL1* and *PlPeL1-like* double deletion mutants) (Supplementary Fig. 2c). Successful deletion mutants were verified by PCR amplification (Supplementary Fig. 2d) and sequencing analysis (Supplementary Fig. 2a–c). Knockout of *PlPeL1* and *PlPeL1-like*, individually or together, did not affect mycelial growth or morphology of *P. litchii* on CJA medium (Supplementary Fig. 3).

To determine the role of these two *PlPeLs* toward *P. litchii* pathogenicity, the abaxial surface of tender litchi leaves were inoculated with zoospore suspensions of WT or knockout mutants, and kept at 25 °C in the dark. We measured the lesion area at 2 days post-inoculation (dpi) and found that the lesions caused by Δ Δ *plpel1/plpel1-like* mutants were significantly smaller than those caused by WT (Fig. 1c, d). However, no significant differences were observed in lesions caused by Δ *plpel1* or Δ *plpel1-like* mutant, compared to WT (Fig. 1c, d). We examined the expression levels of *PlPeL1* and *PlPeL1-like* in Δ *plpel1, plpel1-like* and WT strains at 12 hpi. *PlPeL1-like* was significantly up-regulated in Δ *plpel1* mutants compared to WT, while similarly, *PlPeL1* was significantly up-regulated in Δ *plpel1-like* mutants compared to WT (Supplementary Fig. 4). We deduced that the up-regulation of *PlPeL1* or *PlPeL1-like* in single-gene knockout mutants may serve to compensate for observed virulence defects. Together, these results indicated that *PlPeL1* and *PlPeL1-like* contributed to the virulence of *P. litchii*.

### The enzymatic activity of PlPeL1 and PlPeL1-like is required for virulence

Calcium ion (Ca²⁺) is known to be an essential co-factor for PeL enzymatic activity, and conserved aspartic (D) residues are believed to be the catalytic site of Ca²⁺ binding[15]. Through alignment of 10 PlPeLs proteins from *P. litchii* and homologous proteins of PlPeL1/PlPeL1-like across 15 oomycete species, we identified that both PlPeL1 and PlPeL1-like contained seven conserved aspartic (D) residues (Supplementary Fig. 5, Supplementary Data 1 and Data 2). To probe the functional importance of these conserved D residues, we created mutants PlPeL1^M1 or PlPeL1-like^M1 (both containing SP) by converting them to alanine (A) residues. Following agroinfiltration, we confirmed protein expression through western blot analysis (Supplementary Fig. 6a). We assessed the pectate lyase activity of PlPeL1, PlPeL1-like, and mutants by determining the levels of reducing oligogalacturonic acid using polygalacturonic acid as substrate[40], with RFP as a negative control. These assays demonstrated that the pectate lyase activity of PlPeL1^M1 or PlPeL1-like^M1 was significantly lower than that of PlPeL1 or PlPeL1-like (Fig. 1e).

Next, we expressed PlPeL1, PlPeL1-like and mutants in *N. benthamiana* leaves using agroinfiltration. The infiltrated regions were inoculated with mycelia plugs of *Ph. capsici* at 24 h post-agroinfiltration (hpa). At 2 dpi, we measured the lesion diameter caused by *Ph. capsici* and observed that lesions on leaves expressing PlPeL1 and PlPeL1-like were significantly larger than those on leaves expressing RFP (Fig. 1f, g). In contrast, PlPeL1^M1 or PlPeL1-like^M1 did not significantly affect the infection of *Ph. capsici* on *N. benthamiana* (Fig. 1f, g), suggesting that two mutants lost the ability to promote *N.*

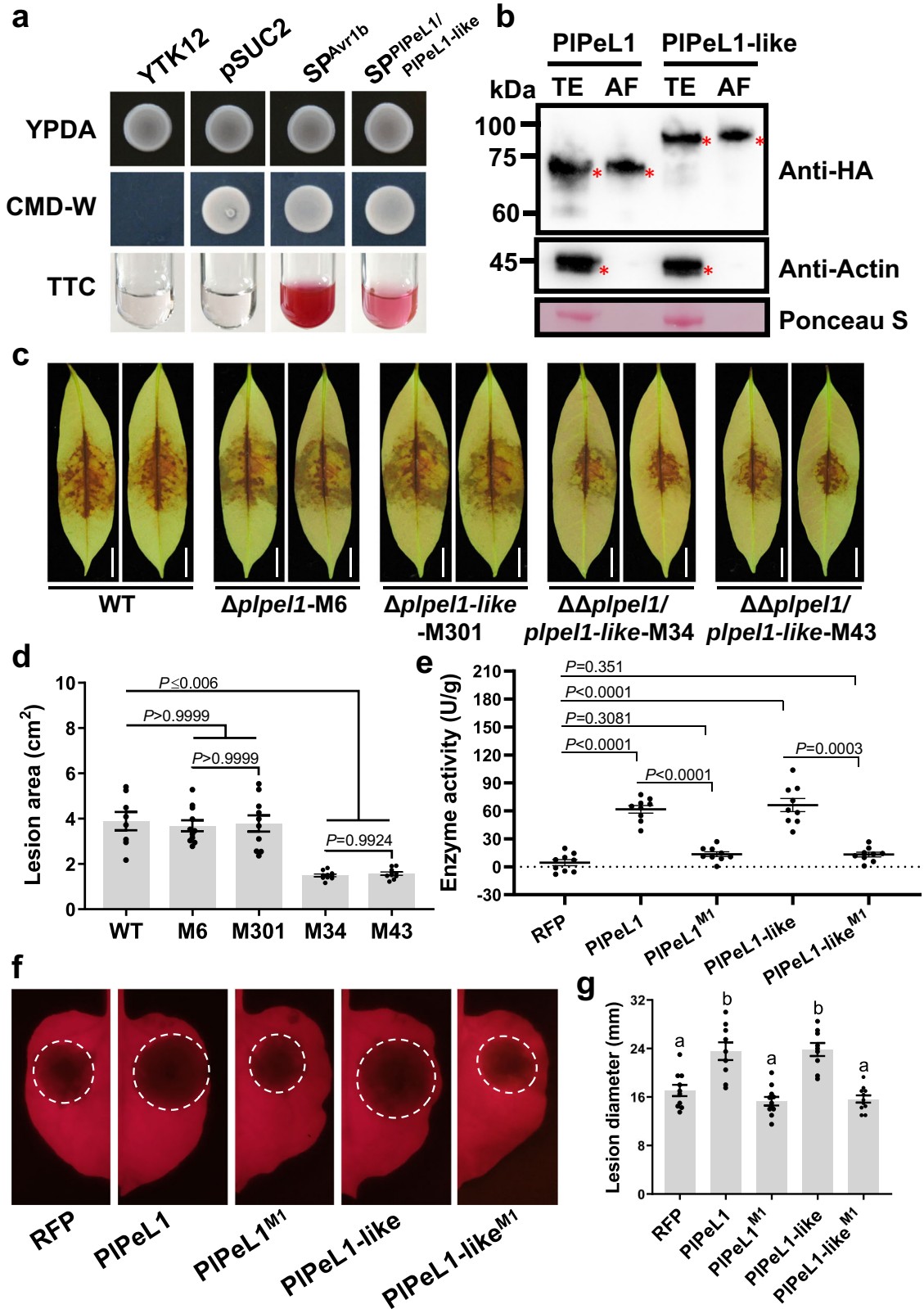

*benthamiana* susceptibility. These results suggested that the enzymatic activity was essential for the virulence function of both PlPeL1 and PlPeL1-like.

### PlPeL1 and PlPeL1-like interact with LcPIP1

We screened interacting proteins of PlPeL/PlPeL1-like by the yeast two-hybrid (Y2H) system using the PlPeL1/PlPeL1-like proteins without SP. By screening the litchi cDNA library, we identified a PlPeL1-interacting protein named PlPeL1-interacting protein 1 (LcPIP1). LcPIP1 is composed of 91 amino acid and possesses a 16 aa SP, but no domain with known function was predicted. Furthermore, the interaction analysis between PlPeL1-like and LcPIP1 revealed that LcPIP1 interacted with both PlPeL1 and PlPeL1-like within the Y2H system (Fig. 2a).

**Fig. 1 | Functional analysis of PlPeL1/PlPeL1-like signal peptide and their role as virulence factors in *Peronophythora litchii*. a** Analysis of the secretion function of PlPeL1 and PlPeL1-like signal peptide (SP). The predicted SP of PlPeL1/PlPeL1-like was cloned into the yeast vector pSUC2. The yeast strain YTK12 carrying the pSUC2 vector can grow on CMD-W medium. SPs of PlPeL1/PlPeL1-like and Avr1b (used as positive control) were fused to mature yeast invertase, converting triphenylte-trazolium chloride (TTC) to red 1,3,5-Triphenylformazan. YTK12 and YTK12 carrying the empty vector were used as negative controls. **b** Western blot was conducted on total tissue extracts (TE) and apoplastic fluid (AF) obtained from *N. benthamiana* leaves after agroinfiltration with PlPeL1-HA or PlPeL1-like-HA for 36 hpa. The extracted proteins were subsequently subjected to immunoblotting using anti-HA or anti-Actin antibodies, as indicated. Red asterisks indicated protein bands of the correct size. Anti-Actin antibody and Ponceau S were used to distinguish between tissue extracts (TE) and apoplastic fluid (AF). **c, d** *PlPeL1* and *PlPeL1-like* double deletion mutants of *P. litchii* showed reduced pathogenicity. Litchi leaves were inoculated with 200 zoospores of WT, Δ*plpel1*, Δ*plpel1-like*, and ΔΔ*plpel1/plpel1-like* mutants at 25 °C in the dark. Inoculated leaves were photographed at 2 dpi, and lesion areas were measured using ImageJ. Images showed representative leaves for

each instance. Scale bars = 1 cm. Data are shown as the mean ± SE (n = 8–12 biologically independent samples). The data were statistically analyzed with one-way ANOVA. **e** PlPeL1/PlPeL1-like or mutants were expressed in *N. benthamiana* leaves using agroinfiltration and pectate lyase activity was analyzed 36 h post-agroinfiltration (hpa). Data are shown as the mean ± SE (n = 9 biologically independent samples). The data were statistically analyzed with one-way ANOVA. **f, g** Expression of PlPeL1 or PlPeL1-like in *Nicotiana benthamiana* enhanced susceptibility to *Phytophthora capsici*. **f** PlPeL1 or PlPeL1-like, PlPeL1^M1 or PlPeL1-like^M1 (seven aspartic residues were changed to alanine residues), or RFP was expressed in *N. benthamiana* leaves by agroinfiltration, then the infiltrated leaves were inoculated with *Ph. capsici* at 24 hpa. Images showed representative leaves for each instance. White circles outline the lesions. **g** Lesion diameters were recorded under ultraviolet (UV) light at 48 h post-infection (hpi). Data are shown as the mean ± SE (n = 9–11 biologically independent samples). Different letters on the graph represent significant differences among samples (One-way ANOVA; *P* < 0.05). All experiments were repeated three times with similar results. Source data are provided as a Source Data file.

We tested the interaction between PlPeL1/PlPeL1-like and LcPIP1/NbPIP1 by Co-IP assays *in planta*, and the results showed that PlPeL1-HA or PlPeL1-like-HA was co-immunoprecipitated with LcPIP1-RFP or NbPIP1-RFP, but not with RFP (Fig. 2b). Furthermore, we performed an in vitro glutathione S-transferase (GST) pull-down assay using proteins produced in *Escherichia coli* strain BL21 strain and found that PlPeL1 or PlPeL1-like interacted with LcPIP1 and NbPIP1 (Fig. 2c). The above results supported the interaction of PlPeL1 and PlPeL1-like with both LcPIP1 and NbPIP1 in vivo and in vitro. Notably, the PlPeL1^M1 or PlPeL1-like^M1 also interacted with LcPIP1 (Supplementary Fig. 6b, c).

### LcPIP1 and its homologs contribute to plant resistance and attenuate the susceptibility induced by PlPeL1/PlPeL1-like

Then, to determine the role of PIP1s in plant immunity to *Ph. capsici*, LcPIP1 was expressed and NbPIP1 was overexpressed in *N. benthamiana* leaves using agroinfiltration, with RFP as control. The expression of LcPIP1 effectively enhanced the *N. benthamiana* resistance to *Ph. capsici*, and so did NbPIP1 (Fig. 3a–c). Next, we silenced *NbPIP1* in *N. benthamiana* using virus-induced gene silencing (VIGS). The transcript level of *NbPIP1* was significantly reduced compared to the control (Fig. 3d). After inoculation with *Ph. capsici*, the lesions in *NbPIP1*-silenced leaves were similar to that in the control (Fig. 3e, f). Furthermore, we generated *atpip1* mutants in *Arabidopsis thaliana* using CRISPR/Cas9-mediated genome editing[41]. Single nucleotide insertions or a 10-nucleotide deletion in *AtPIP1* exons resulted in premature stop codons, leading to loss of *AtPIP1* function (Supplementary Fig. 7a). The *atpip1* mutants did not show remarkable morphology alterations compared with the Col-0 (Supplementary Fig. 7b), but they were more susceptible to *Ph. capsici* compared to the control (Col-0) (Fig. 3g–i). In addition, *AtPIP1* exhibited up-regulation in the Col-0 plants during *Ph. capsici* infection (Supplementary Fig. 7c). These results suggested that LcPIP1 and its homologs contributed to plant immunity to *Ph. capsici*.

To study the interaction between PlPeL1/PlPeL1-like and PIP1s *in planta*, LcPIP1 was co-expressed with PlPeL1, PlPeL1-like or RFP in *N. benthamiana* leaves. When PlPeL1 was co-expressed with LcPIP1, the *N. benthamiana* leaves showed significantly smaller disease lesions than those lesion on leaves co-expressing PlPeL1 and RFP (Fig. 3a–c). Furthermore, *N. benthamiana* leaves co-expressing LcPIP1 and PlPeL1 showed similar resistance to *Ph. capsici* compared with leaves co-expressing LcPIP1 and RFP. The co-expression of PlPeL1-like and LcPIP1 also had similar results (Fig. 3a–c). Western blot analysis confirmed the expression of the above proteins in *N. benthamiana* (Supplementary Fig. 8a). The co-expression of LcPIP1 with PlPeL1 or PlPeL1-like did not interfere with the pectate lyase activity of PlPeL1 and PlPeL1-like (Supplementary Fig. 8b). Next, we analyzed the function of PlPeL1/PlPeL1-like in *NbPIP1*-silenced plants; PlPeL1, PlPeL1-like, and RFP were

expressed in TRV-*NbPIP1* plants through agroinfiltration, which were then inoculated with *Ph. capsici*. The lesions of *Ph. capsici* on PlPeL1-expressing *NbPIP1*-silenced leaves were larger than the lesions in the control plants. However, no differences in lesion diameter were found between TRV-*NbPIP1* and TRV-*GUS* plants which expressed PlPeL1-like and RFP (Fig. 3j, k). The above results confirmed that PIP1s attenuated the susceptibility of *N. benthamiana* induced by PlPeL1 and PlPeL1-like.

### LcPIP1 activates various immune responses in *N. benthamiana* and its homologs are widely present in land plants

The expression profile of the *LcPIP1* was determined by qRT-PCR. *LcPIP1* was up-regulated during *P. litchii* infection, especially at 6, 12, and 24 hpi (Fig. 4a), suggesting that *LcPIP1* expression is responsive to *P. litchii*. Next, we compared the transcription levels of *LcPIP1* in litchi infected with the WT strain to that in the knockout mutants (M6, M301, M34, and M43). In comparison to WT-infected litchi, the transcription levels of *LcPIP1* were less up-regulated at 6 and 12 hpi than in litchi inoculated with double knockout mutants-infected (Fig. 4b). These results indicated that *LcPIP1* expression might be indirectly regulated by PlPeL1 and PlPeL1-like in litchi. Furthermore, *LcPIP1* exhibited ubiquitous expression in various litchi tissues, including leaves, branches shoots, panicles, flowers, and fruitlets (Supplementary Fig. 9a).

To further investigate the mechanisms underlying PIP1-mediated resistance, we expressed LcPIP1 and evaluated its role in inducing cell death. Interestingly, LcPIP1 induced cell death at 3 dpa, accompanied by ROS accumulation and callose deposition at 36 hpa in *N. benthamiana*, while the negative control RFP did not (Fig. 4c, d; Supplementary Fig. 9b, c). Moreover, co-expressing PlPeL1^M1/PlPeL1-like^M1 with LcPIP1 resulted in increased electrolyte leakage in the injection area compared to the control (co-expression of RFP and LcPIP1) (Fig. 4e), suggesting that PlPeL1^M1/PlPeL1-like^M1 could enhance LcPIP1-induced cell death. Western blot analysis showed that all the proteins were expressed in *N. benthamiana* (Supplementary Fig. 9d). Furthermore, we examined the expression levels of genes related to salicylic acid (SA), jasmonic acid (JA), and the ethylene signaling pathway, and defense-related genes *NbRbohb*, in LcPIP1- or RFP-expressing leaves at different time points (24, 36, and 48 h). We found that *NbPR1*, *NbPR2*, *NbNDR1*, *NbLOX*, and *NbRbohb* were significantly up-regulated in the LcPIP1-expressing leaves from 24 to 48 h, compared with the control. *NbERF1* was up-regulated at 24 h after expression of LcPIP1 (Fig. 4f). Therefore, LcPIP1 may induce a *N. benthamiana* immune response by activating SA- and JA- mediated defense pathways. In addition, we examined the expression levels of some PTI marker genes and the results showed that *NbPTI5*, *NbAcre31*, *NbWRKY7*, and *NbWRKY8* were significantly up-regulated in the LcPIP1-expressing leaves. *NbCYP71D20* was up-regulated at 24 hpa, and subsequently down-regulated at

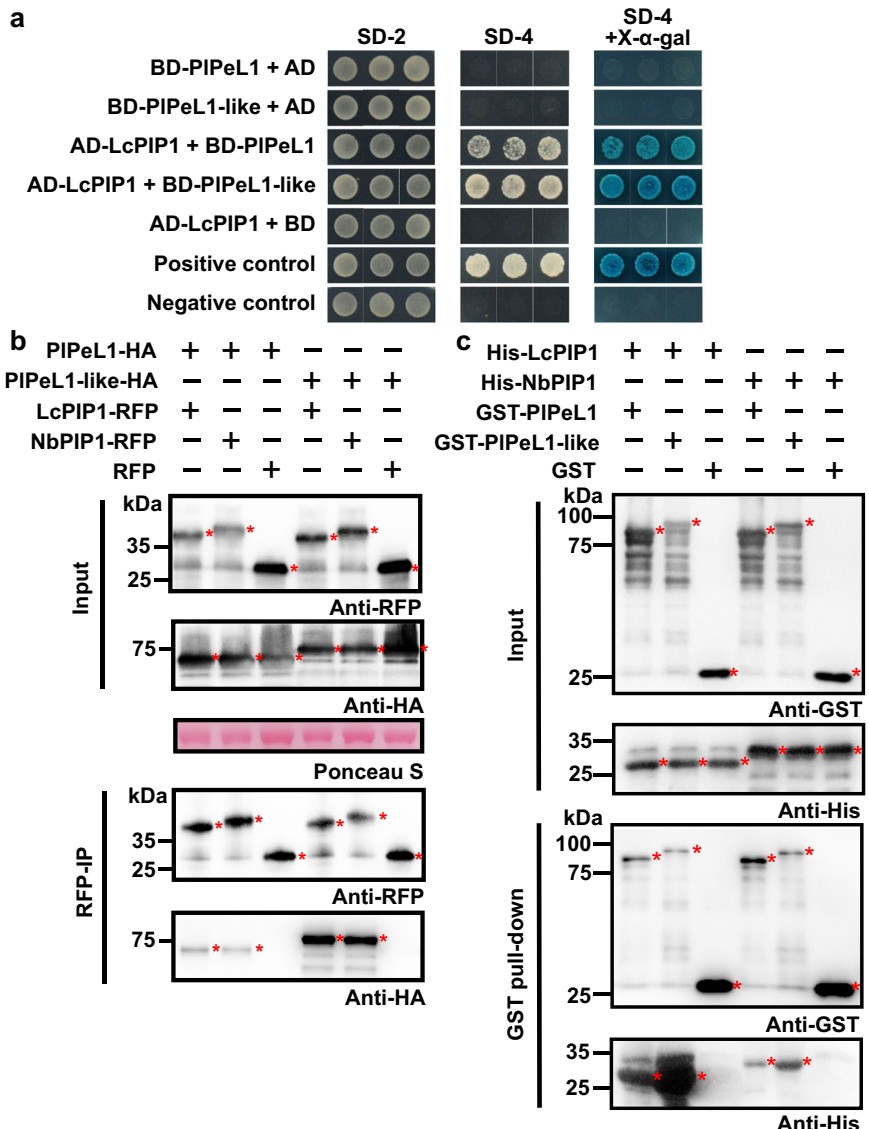

**Fig. 2 | PlPeL1 and PlPeL1-like interact with LcPIP1. a** Yeast two-hybrid (Y2H) assays demonstrated that LcPIP1 interacted with PlPeL1 and PlPeL1-like. *PlPeL1* and *PlPeL1-like* were cloned into pGBKT7, and *LcPIP1* was cloned into pGADT7 vector. AD, pGADT7 vector; BD, pGBKT7 vector. A combination of BD-53 and AD-T was used as a positive control, and a combination of BD-Lam and AD-T was used as a negative control. Yeast transformants were grown on SD/-Trp/-Leu or SD/-Trp/-Leu/-His/-Ade selective medium with X-a-gal. Images were taken 3 days post-inoculation. **b** LcPIP1 interacted with PlPeL1 and PlPeL1-like *in planta*. PlPeL1-HA and PlPeL1-like-HA were co-expressed with LcPIP1-RFP, NbPIP1-RFP, or RFP in *N.* *benthamiana* leaves. Protein complexes were immunoprecipitated with RFP-Trap-M beads. Co-precipitation was detected by western blot. **c** LcPIP1 physically interacted with PlPeL1 and PlPeL1-like in vitro. GST-PlPeL1-, GST-PlPeL1-like-, or GST-bound beads were incubated with bacteria lysate containing His-LcPIP1 or His-NbPIP1. Co-precipitation was detected by western blot. Red asterisks indicated protein bands of the correct size. Ponceau S staining of Rubisco was used to indicate loading quantity of protein samples. All experiments were repeated three times with similar results.

36–48 hpa (Fig. 4g). Overall, these results suggested that LcPIP1 can activate diverse defense responses in *N. benthamiana*.

To assess the presence of LcPIP1 homologs in plants, we conducted a BLASTP search to obtain the homologs of LcPIP1 from 23 different land plant species (Supplementary Data 3). Additionally, litchi possesses a paralogous protein of LcPIP1, named LcPIP1PP, with 46% identity. All these proteins carry an SP and possess a conserved "VDMASG" motif. However, the homologous protein PpPIP1 in *Prunus persica* contains a slightly different motif, "VDMGSG" (Supplementary Data 3 and Supplementary Fig. 10a). Phylogenetic analysis indicated that LcPIP1 was most closely related to homologs from Sapindaceae, consistent with the established evolutionary relationships among these plant species (Fig. 5a). We expressed LcPIP1 and 11 homologs of LcPIP1 in *N. benthamiana*, and confirmed their expression through

western blot analysis (Supplementary Fig. 10b). Interestingly, expression of the all tested LcPIP1 homologs from different groups, induced visible cell death at 4 dpa (Fig. 5b). These data indicated that sequence homogeneity of PIP1s is conserved in land plants, along with the cell-death-inducing function.

### Signal peptide and the VDMASG motif are critical for LcPIP1 cell death-inducing activity

The function of the SP of LcPIP1 was evaluated using a yeast secretion assay[37] (Supplementary Fig. 11a). After agroinfiltration, the LcPIP1 protein was detected in the apoplastic fluid (AF) and total tissue extracts (TE) (Fig. 5c). Furthermore, we expressed LcPIP1 and two mutants, LcPIP1$^{\Delta SP}$ (lacking its SP) and PR1$^{SP}$-LcPIP1$^{\Delta SP}$ (the SP of pathogenesis-related protein 1 (NbPR1) fused with LcPIP1$^{\Delta SP}$), in *N.*

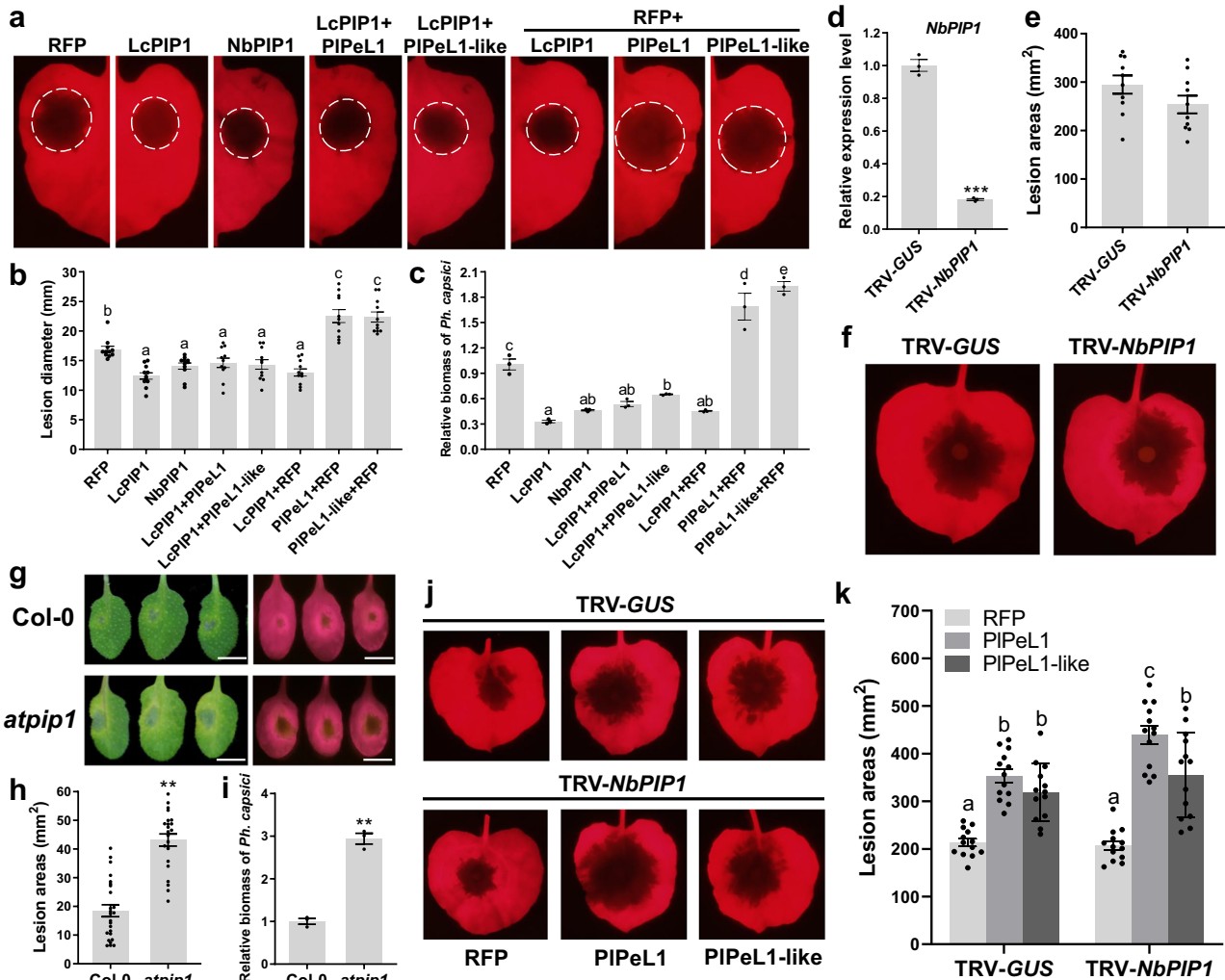

**Fig. 3 | LcPIP1 and homologs enhance plants resistance and attenuate susceptibility induced by PlPeL1/PlPeL1-like. a** Representative photographs of *N. benthamiana* leaves inoculated with *Ph. capsici*. Leaves expressing LcPIP1, NbPIP1, RFP, LcPIP1 with PlPeL1 or PlPeL1-like, RFP with LcPIP1, PlPeL1 or PlPeL1-like were inoculated with *Ph. capsici* at 24 hpa. Lesions were photographed under ultraviolet (UV) light at 2 dpi. White circles outline the lesions. **b** Lesion diameters were recorded under ultraviolet (UV) light. Data are shown as the mean ± SE (n = 11 biologically independent samples). Different letters on the graph represent significant differences among samples (One-way ANOVA; *P* < 0.05). **c** DNA from *Ph. capsici* infected regions was isolated and relative *Ph. capsici* biomass was measured to evaluate the severity of infection by qPCR. Data are shown as the mean ± SE of three replicates. Different letters on the graph represent significant differences among samples (One-way ANOVA; *P* < 0.05). **d** qRT-PCR analysis relative expression of *NbPIP1* in *NbPIP1*- or *GUS*- silenced plant leaves. *NbEF1α* was used as the endogenous control. Data are shown as the mean ± SE of three replicates. Asterisks represent significant differences (***P* < 0.001) based on Two-tailed Student's *t*-test. **e, f** Detached leaves of VIGS plants inoculated with *Ph. capsici* and the lesion areas

were measured using ImageJ at 2 dpi under ultraviolet (UV) light. Data are shown as the mean ± SE (n = 10 biologically independent samples). Two-tailed Student's *t*-test was used for significance analysis. **g-i** The Arabidopsis *atpip1* mutants exhibited reduced resistance to *Ph. capsici*. Leaves from the indicated plants were inoculated with *Ph. capsici* zoospores. **g** Disease symptoms were photographed under white or UV light at 36 hpi. Scale bars = 1 cm. **h** Lesion areas on Arabidopsis leaves caused by *Ph. capsici* (n = 23–25 biologically independent samples). Two-tailed Student's *t*-test was used for significance analysis. **i** Quantification of *Ph. capsici* biomass by qPCR analysis to measure the ratios of *Ph. capsici* to *N. benthamiana* DNA. Data are shown as the mean ± SE of three replicates. Two-tailed Student's *t*-test was used for significance analysis. **j, k** The RFP, PlPeL1, or PlPeL1-like -expressing VIGS plant leaves, were inoculated with mycelia plugs of *Ph. capsici*, and the lesion areas were measured using ImageJ at 2 dpi. Data are shown as the mean ± SE (n = 13 biologically independent samples). Different letters on the graph represent significant differences among samples (One-way ANOVA; *P* < 0.05). All experiments were repeated three times with similar results. Source data are provided as a Source Data file.

*benthamiana*. Western blot analysis showed that all the proteins were expressed in *N. benthamiana* (Supplementary Fig. 11b). Notably, expression of LcPIP1 or PR1^SP^-LcPIP1^ΔSP^ induced cell death at 3 dpa, while LcPIP1^ΔSP^ and the negative control RFP could not induce cell death in *N. benthamiana* (Fig. 5d). In addition, the previous studies indicated that PsXEG1 targeted to the extracellular space of *N. benthamiana* tissue induces cell death[42]. Expression of LcPIP1^SP^-PsXEG1^ΔSP^ also induced cell death, while PsXEG1^ΔSP^ and LcPIP1^SP^-RFP did not (Supplementary Fig. 11c). Western blot analysis confirmed the expression of all proteins in *N. benthamiana* (Supplementary Fig. 11d).

All the above results demonstrated that the signal peptide possessed secretion function and was required for LcPIP1-inducing cell death.

To identify the key residues required for LcPIP1-inducing cell death, six truncated or site-directed mutants were constructed and expressed in *N. benthamiana* leaves (Fig. 5e). Western blot analysis showed that all the proteins were expressed in *N. benthamiana* (Supplementary Fig. 11e). The truncated protein M1 (LcPIP1^1-75 aa^) induced cell death like the full-length LcPIP1 protein. Truncated proteins M2 and M4 (LcPIP1^1-69 aa^ and LcPIP1^1-38 aa^) induced cell death and ROS accumulation in *N. benthamiana* to a much slighter degree

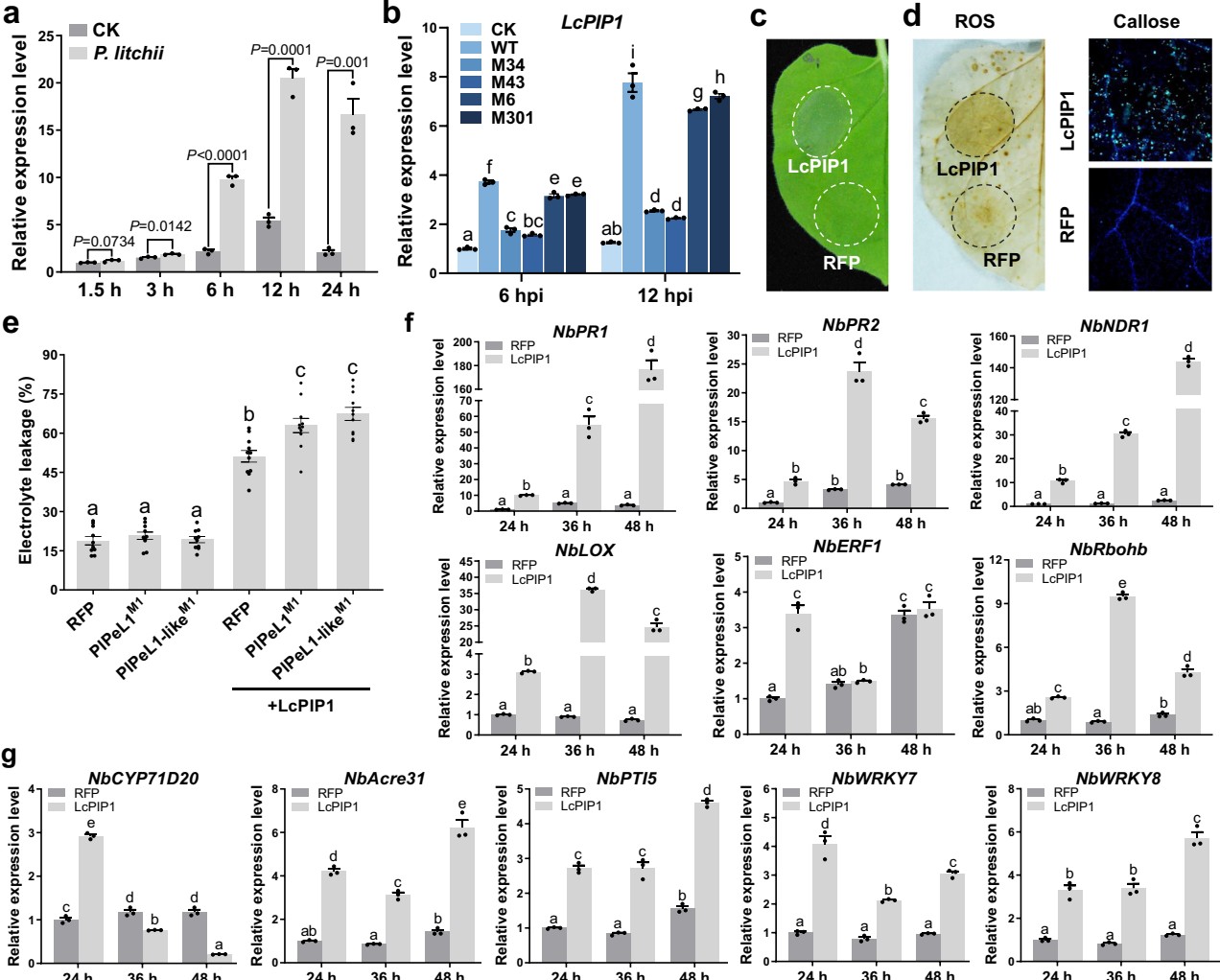

**Fig. 4 | LcPIP1 induces plant immunity responses in *Nicotiana benthamiana*.**
**a** qRT-PCR analysis of *LcPIP1* expression in response to *Peronophythora litchii*. Total RNAs were extracted from the litchi leaves inoculated with *P. litchii* or water (CK) at different time points (0–24 h). The expression level of *LcPIP1* in leaves inoculated with water at 0 h was set at 1. *LcActin* was used as the endogenous control. Data are shown as the mean ± SE of three replicates. Two-tailed Student's *t*-test was used for significance analysis. **b** Total RNAs were extracted from the litchi leaves inoculated with water (CK), wild-type (WT), ΔΔ*plpel1/plpel1-like* (M34 and M43), Δ*plpel1* (M6), or Δ*plpel1-like* (M301) at 6 or 12 hpi. The expression level of *LcPIP1* in leaves inoculated with water at 6 hpi was set at 1. Data are shown as the mean ± SE of three replicates. Different letters on the graph represent significant differences among samples (One-way ANOVA; *P* < 0.05). **c** *N. benthamiana* leaves were infiltrated with *Agrobacterium tumefaciens* strains carrying C-terminal HA-tagged LcPIP1 or RFP constructs. Photographs were taken at 4 dpa. RFP was used as negative control. **d** Accumulation of ROS

and callose deposition in *N. benthamiana* leaves expressing LcPIP1 at 2 dpa and RFP expression served as a control. **e** Quantification of cell death by measuring electrolyte leakage at 2 dpa. Electrolyte leakage from infiltrated leaf discs was measured as a percentage of leakage from boiled discs. Data are shown as the mean ± SE (n = 11 biologically independent samples). Different letters on the graph represent significant differences among samples (One-way ANOVA; *P* < 0.05). **f, g** qRT-PCR analyzed relative expression of defense-related marker genes and pathogen-associated molecular pattern (PAMP)-triggered immunity (PTI) marker genes in *N. benthamiana* leaves expressing LcPIP1 and RFP at 24, 36, and 48 h post-agroinfiltration. The expression level of genes in leaves expressing RFP at 24 hpa was set at 1. *NbEF1α* was used as the endogenous control. Data are shown as the mean ± SE of three replicates. Different letters on the graph represent significant differences among samples (One-way ANOVA; *P* < 0.05). All experiments were repeated three times with similar results. Source data are provided as a Source Data file.

than that triggered by the full-length LcPIP1 (Fig. 5e). Significantly, M5 (LcPIP1^1-34 aa) completely lost the ability to trigger cell death, while M6 (LcPIP1^1-34&VDMASG aa), which was fused M5 the "VDMASG" motif, induced cell death and ROS accumulation in *N. benthamiana* (Fig. 5e). Moreover, site-directed mutant M3, in which the conserved "VDMASG" motif was changed to six alanine residues, induced weak cell death (Fig. 5e). These findings suggested that the C-terminal "VDMASG" motif was important for the cell death-inducing activity of LcPIP1.

## SERK3 is required for LcPIP1-induced cell death
To dissect the immune response mechanism induced by LcPIP1, we conducted VIGS-mediated silencing of key components in the

plant innate immunity signaling pathway in *N. benthamiana*. These genes included *SERK3A/B*, *SOBIR1*, *EDS1*, *NDR1*, *NPR1*, *WRKY1*, *WRKY3*, *SGT1*, *MEK2*, *MAPK3α*, *SIPK*, and *RAR1*. The relative expression of these genes was verified by qRT-PCR (Supplementary Fig. 12a). Subsequently, cell death was assessed following LcPIP1 expression in these silenced plants. LcPIP1-induced cell death was greatly compromised in *NbSERK3*-silenced plants compared to *GUS*-silenced (Supplementary Fig. 12b, c). Western blot analysis confirmed that LcPIP1 was successfully expressed in *NbSERK3*-silenced plants (Supplementary Fig. 12d). Furthermore, the cell death proportion in the other gene-silenced plants was comparable to the control, but slightly attenuated in plants silenced for *MEK1* (Supplementary Fig. 12b, c). Furthermore,

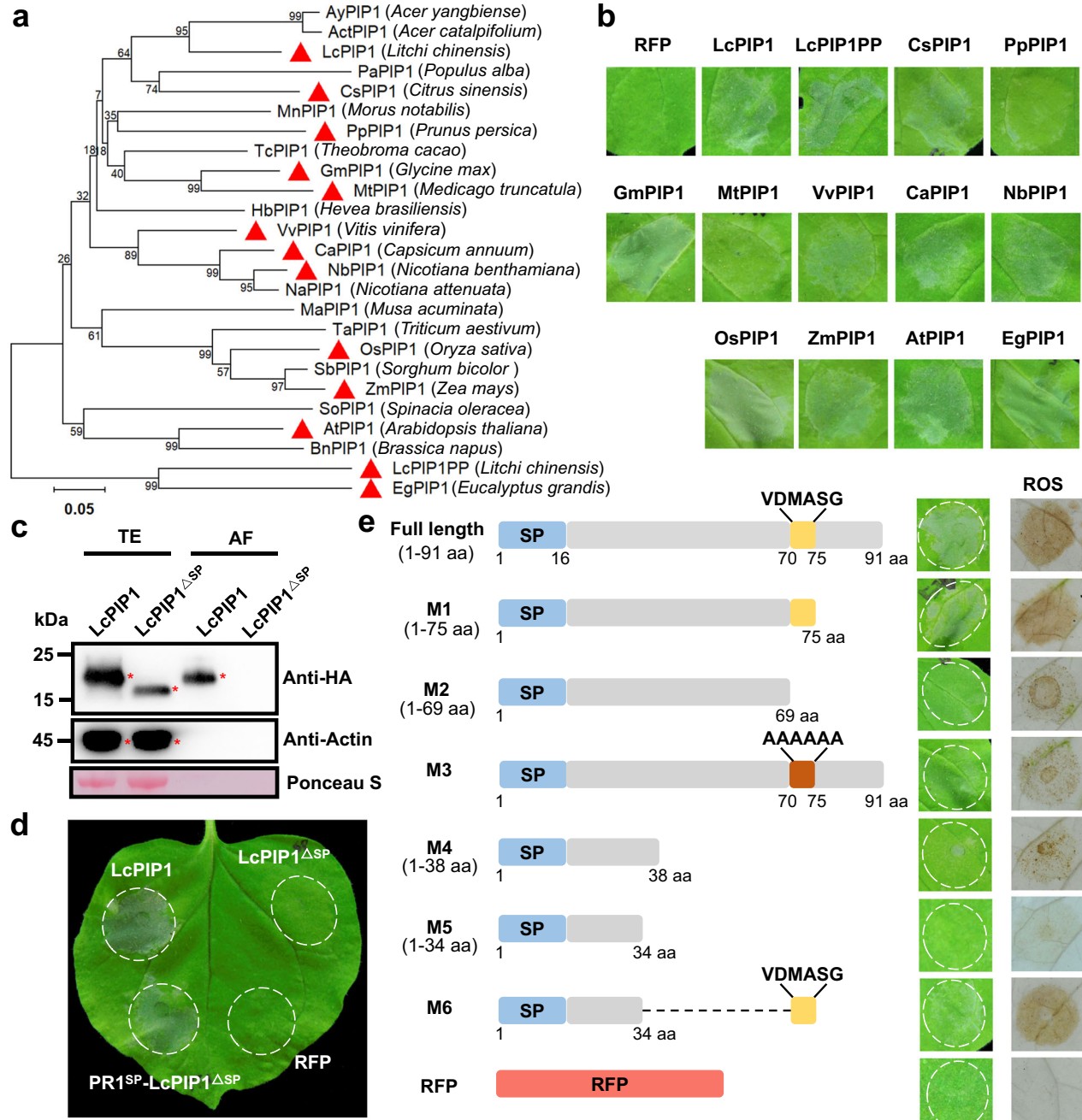

**Fig. 5 | LcPIP1 homologs are widely present in land plants and the VDMASG motif is critical for cell death-inducing activity. a** Phylogenetic analysis illustrates the relationship among LcPIP1, its closest paralogous protein (LcPIP1PP), and 23 homologs from diverse species. The tree was constructed using the neighbor-joining algorithm. The proteins that induced cell death in *N. benthamiana* are indicated by red triangles. **b** Cell death assays of LcPIP1 homologs expressed in *N. benthamiana* leaves. Representative *N. benthamiana* leaves were photographed at 4 dpa. **c** Western blot analysis of total tissue extracts (TE) and apoplastic fluid (AF) from *N. benthamiana* leaves agroinfiltrated with LcPIP1-HA or LcPIP1$^{\Delta SP}$-HA for 36 hpa. Extracted proteins were analyzed using immunoblotting with anti-HA or anti-Actin antibodies, as indicated. Red asterisks indicated protein bands of the correct size. Ponceau S staining of Rubisco was used to indicate loading quantity of protein samples. **d** LcPIP1$^{\Delta SP}$ was unable to induce cell death, but the SP of PR1 restored the ability of the LcPIP1 deletion mutant to trigger cell death. Photographs were taken at 4 dpa. **e** Schematic diagrams of the protein structures of six truncated and site-directed mutants of LcPIP1 were shown on the left. Cell death symptoms and accumulation of ROS in *N. benthamiana* leaves expressing full-length and mutants were shown on the right. Photographs were taken at 4 dpa. Accumulation of ROS was observed in protein-expressing leaves at 2 dpa. These experiments were repeated three times with similar results.

compared with TRV-*GUS* plants, silencing of *NbSERK3* compromised LcPIP1-induced transcript abundance of *NbNDR1*, *NbWRKY7*, and *NbWRKY8* (Supplementary Fig. 12e). Transcript abundance of the three genes was not different in *NbSERK3*- or *GUS*- silenced plants expressing RFP (Supplementary Fig. 12e).

The receptor-like kinase SERK3 (also known as BAK1) is the central regulator of innate immunity in plants[34]. To further validate that the significance of NbSERK3 in mediating LcPIP1-triggered immune responses in *N. benthamiana*, we expressed LcPIP1 or INF1 in the *BAK1*-knockout *N. benthamiana*[43,44]. INF1 and its homologs

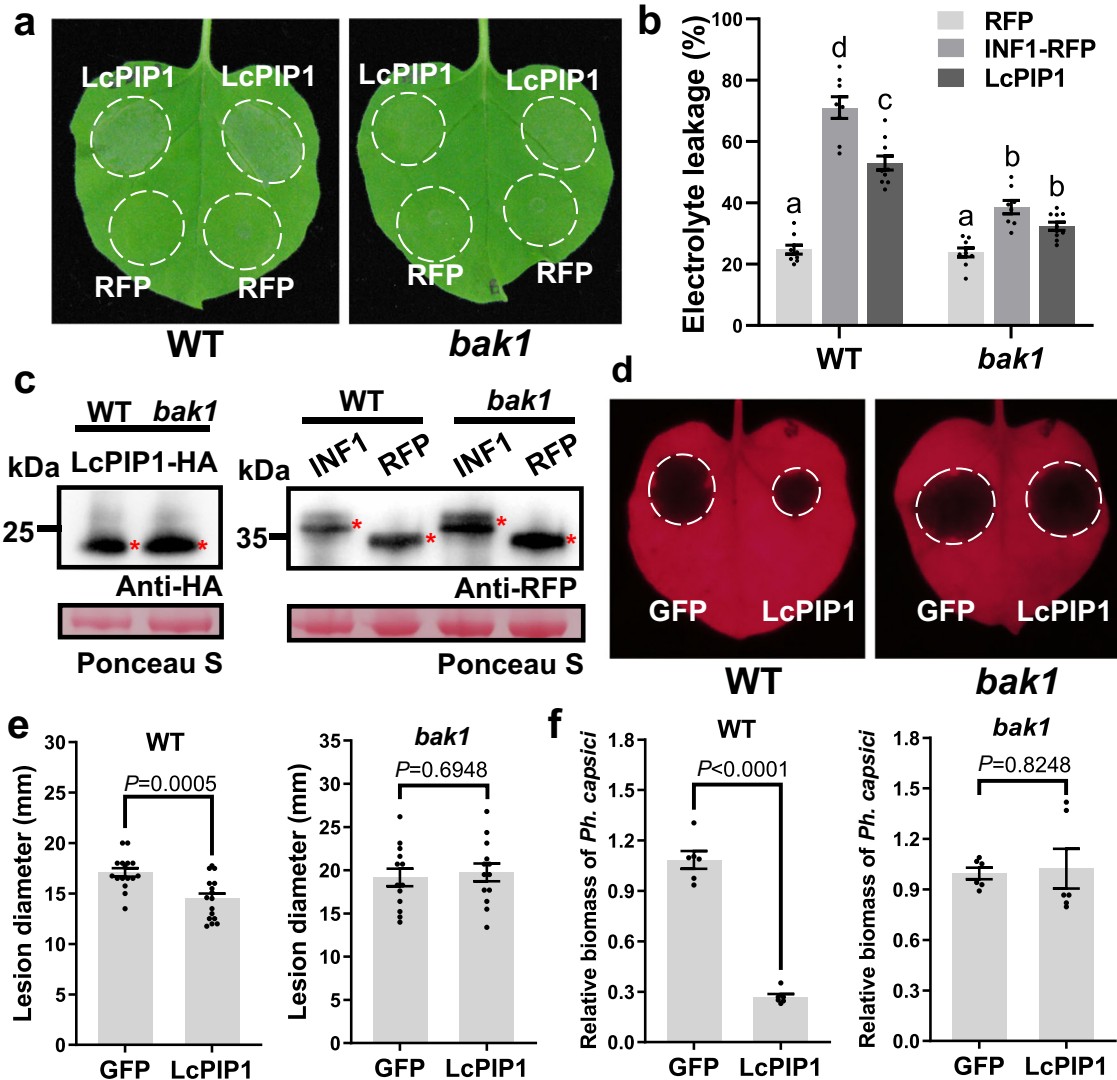

**Fig. 6 | SERK3 is required for LcPIP1-induced cell death. a** Cell death triggered by LcPIP1 in wild-type (WT) and *BAK1*-knockout lines. *N. benthamiana* leaves were infiltrated with *Agrobacterium tumefaciens* strains carrying C-terminal HA-tagged LcPIP1. Photographs were taken at 4 dpa. **b** LcPIP1-HA, INF1-RFP, and RFP (as a vector control for INF1-RFP) were expressed in both WT and *bak1* plants. Quantification of cell death by measuring electrolyte leakage at 2 dpa. Electrolyte leakage from infiltrated leaf discs was measured as a percentage of leakage from boiled discs. Data are shown as the mean ± SE (n = 8–10 biologically independent samples). Different letters on the graph represent significant differences among samples (One-way ANOVA; *P* < 0.05). **c** Protein expression of LcPIP1-HA, INF1-RFP, and RFP was verified by western blot. Red asterisks indicated protein bands of the

correct size. Ponceau S staining of Rubisco was used to indicate loading quantity of protein samples. **d, e** The GFP- or LcPIP1- expressing WT and *BAK1*-knockout lines, were inoculated with mycelia plugs of *Ph. capsici*, and the lesion diameters were measured at 48 hpi. Data are shown as the mean ± SE (n = 13–16 biologically independent samples). Two-tailed Student's *t*-test was used for significance analysis. **f** DNA from *Ph. capsici* infected regions was isolated and relative *Ph. capsici* biomass was measured to evaluate the severity of infection by qPCR. Data are shown as the mean ± SE of six replicates. Two-tailed Student's *t*-test was used for significance analysis. These experiments were repeated three times with similar results. Source data are provided as a Source Data file.

exhibit high conservation in oomycete-specific, making them well-characterized canonical ELI-1 elicitors capable of inducing cell death[43,45]. The extent of cell death was measured by monitoring electrolyte leakage. Compared with the wild-type control, LcPIP1 or INF1 induced cell death was reduced in *BAK1*-knockout lines at 2 dpa (Fig. 6a, b). Western blot analysis confirmed that proteins were successfully expressed (Fig. 6c). Next, we analyzed the susceptibility of *BAK1*-knockout and the wild-type *N. benthamiana* expressing *LcPIP1* or *GFP*. In the wild-type, LcPIP1 expression resulted in smaller lesions caused by *Ph. capsici*. However, in the *BAK1*-knockout plants, the expression of LcPIP1 did not enhance resistance against *Ph. capsici* (Fig. 6d–f). Overall, these results demonstrated that NbSERK3 (BAK1) was a key regulator in LcPIP1-triggered immune responses in *N. benthamiana*.

## LcPIP1 interacts with SERK3 proteins

We successfully cloned four *NbSERK3* (*BAK1*) homologs based on NbSERK3 sequences from NCBI and the Sol Genomics Network database, including NbSERK3A (Niben101Scf11779g01002.1), NbSERK3B (Niben101Scf00279g02022.1), NbBAK1a (Niben101Scf02128g00022.1), and NbBAK1b (Niben101Scf02513g11004.1). Co-IP assays analyzed the interaction between LcPIP1 and four NbSERK3s in vivo. LcPIP1-Flag co-immunoprecipitated with NbSERK3A-RFP, NbSERK3B-RFP, NbBAK1a-RFP, and NbBAK1b-RFP, but not RFP (Fig. 7a). To further validate the relationship between LcPIP1 and SERK3 of litchi, *LcSERK3* (*LITCHI011582*), the gene encoding the protein homologous to NbSERK3A, was cloned from litchi cDNA and subsequently constructed into plasmids pBinRFP and pGEX-6P-1 for Co-IP and GST pull-down assays. Both Co-IP and GST pull-down assays confirmed that

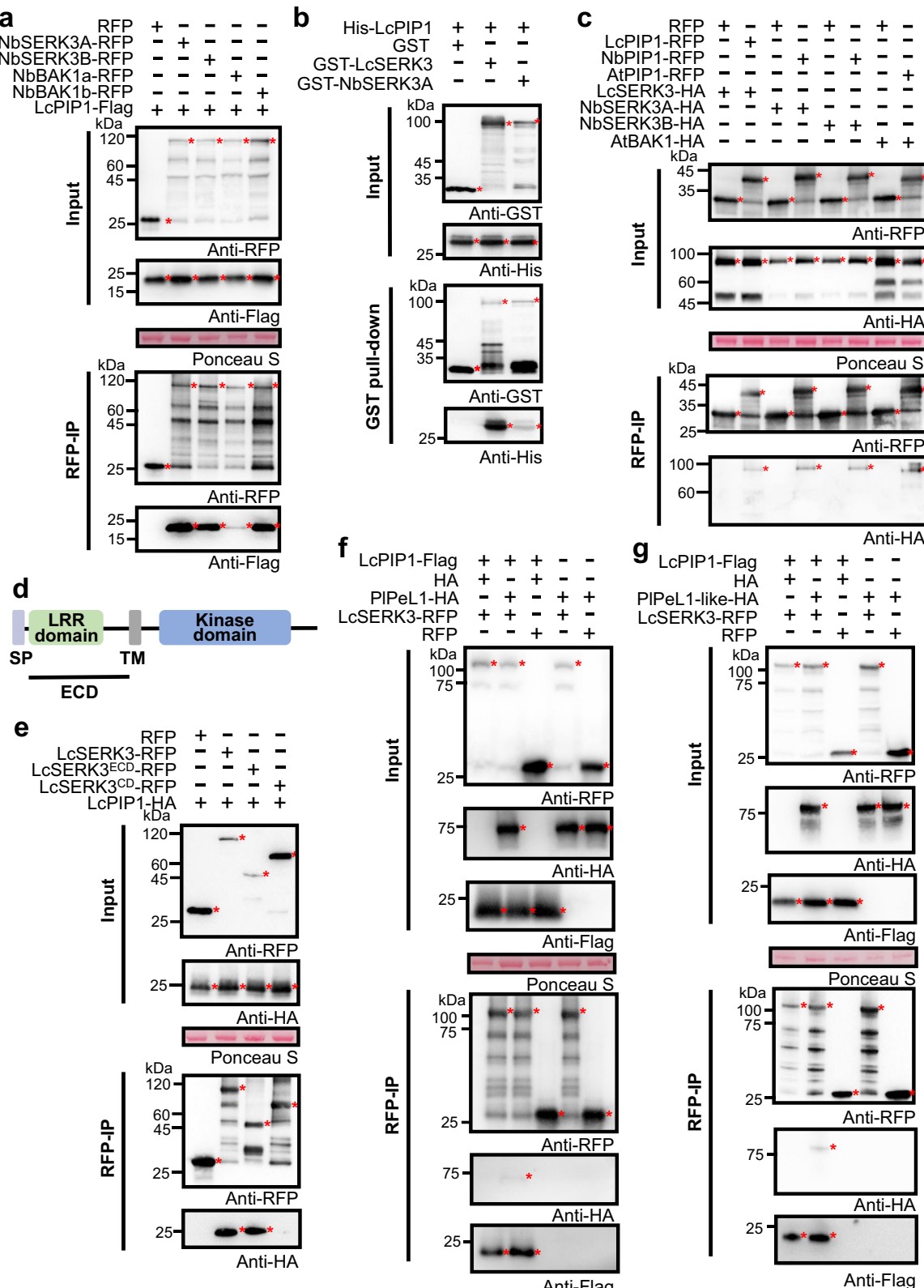

LcPIP1 interacted with LcSERK3 (Fig. 7b and Supplementary Fig. 13a). In addition, we independently validated the interactions between NbPIP1 and NbSERK3A/B, as well as AtPIP1 and AtBAK1 using Co-IP assays (Fig. 7c and Supplementary Fig. 13b).

LcSERK3 possesses both an extracellular domain (ECD) and a cytoplasmic domain (CD) (Fig. 7d). We performed Co-IP assays to investigate the interaction between LcPIP1 and LcSERK3 and identify

the domain responsible for their association. LcPIP1-HA was co-expressed with LcSERK3-RFP, LcSERK3^ECD-RFP, LcSERK3^CD-RFP, or RFP in *N. benthamiana*. Co-IP assays showed that the extracellular domain of LcSERK3 interacted with LcPIP1, but the cytoplasmic domain did not (Fig. 7e).

We next tested whether the LcPIP1-LcSERK3 complex could be disrupted by PlPeL1 or PlPeL1-like. Co-IP assays between LcPIP1-Flag

**Fig. 7 | PIP1s interact with SERK3 (BAK1), and PlPeL1/PlPeL1-like do not disrupt the LcPIP1-LcSERK3 interaction. a** LcPIP1-Flag was co-expressed with NbSERK3A-RFP, NbSERK3B-RFP, NbBAK1a-RFP, NbBAK1b-RFP, or RFP in *N. benthamiana* leaves. Protein complexes were immunoprecipitated with RFP-Trap-M beads. Co-precipitation was detected by western blot. **b** LcPIP1 interacted with LcSERK3 and NbSERK3A in vitro. GST-LcSERK3-, GST-NbSERK3A-, or GST- bound beads were incubated with bacteria lysate containing His-LcPIP1. Co-precipitation was detected by western blot. **c** PIP1s interacted with SERK3s *in planta*. SERK3 (BAK1) was co-expressed with PIP1-RFP or RFP in *N. benthamiana* leaves. Protein complexes were immunoprecipitated with RFP-Trap-M beads and detected by western blot. **d** Schematic representation of LcSERK3, featuring its extracellular leucine-rich repeat (LRR) domain in green, and intracellular kinase domain in blue.

Transmembrane (TM) segment in gray and signal peptide (SP) in purple. **e** LcPIP1-HA was co-expressed with LcSERK3-RFP, LcSERK3$^{ECD}$-RFP, LcSERK3$^{CD}$-RFP, or RFP in *N. benthamiana* leaves. Protein complexes were immunoprecipitated with RFP-Trap-M beads. Co-precipitation was detected by western blot. **f, g** PlPeL1/PlPeL1-like did not disrupt the LcPIP1-LcSERK3 interaction. HA-tagged PlPeL1 or PlPeL1-like was co-expressed with Flag-tagged LcPIP1 and RFP-tagged LcSERK3 in *N. benthamiana*. Protein complexes were immunoprecipitated with RFP-Trap-M beads. Co-precipitation was detected by western blot. Red asterisks indicated protein bands of the correct size. Ponceau S staining of Rubisco was used to indicate loading quantity of protein samples. These experiments were repeated three times with similar results.

and LcSERK3-RFP were performed in *N. benthamiana* leaves expressing PlPeL1-HA or HA. We observed that LcPIP1-LcSERK3 interactions were not affected by the presence of PlPeL1 (Fig. 7f). The co-expression of PlPeL1-like, LcPIP1, and LcSERK3 also had similar results (Fig. 7g). These results suggested that PlPeL1 or PlPeL1-like did not disrupt the LcPIP1-LcSERK3 interaction. In addition, the LcPIP1$^{M3}$ with "VDMASG" motif changed to "AAAAAA" and with compromised cell death-inducing capacity, also could interact with LcSERK3 as shown by the Co-IP assay, to a comparable level as the full-length LcPIP1 protein (Supplementary Fig. 14).

## Silencing of *NbPIP1* in *N. benthamiana* compromises INF1-triggered cell death

Cell death triggered by INF1 requires NbSERK3[46], so we tested whether *NbPIP1* was associated with INF1-induced cell death. The experimental results showed that silencing of *NbPIP1* or *NbSERK3* delayed INF1-mediated HR development. INF1-induced cell death was strong in the TRV-*GUS* plants at 48 h after transient expression of INF1, but only slight in *NbPIP1*-silenced plants (Fig. 8a–c). The expression of INF1 was analyzed by western blot which revealed that delayed cell death was not caused by different protein levels (Fig. 8a). We examined some defense-related genes and PTI marker genes in TRV-*GUS* and TRV-*NbPIP1* plants after INF1 expression. The transcription leaves of nine marker genes other than *WRKY8* were lower in TRV-*NbPIP1* compared to the control (Fig. 8d). Moreover, *NbPIP1* was significantly up-regulated in the INF1-expressing leaves from 24 to 36 h compared with the control (Fig. 8e), again indicating that *NbPIP1* was involved in the INF1-mediated response. However, upon flg22 treatment, the expression of *FRK* and *WRKY33* in the *NbPIP1*-silenced plants was comparable to that in the control plant (Supplementary Fig. 15). These results indicated that NbPIP1 positively regulated *N. benthamiana* PTI responses induced by INF1, but not by flg22.

Taken together, LcPIP1 was found to be a cell death-inducing protein which positively regulated plant defense (Fig. 8f). PIP1 induced SERK3- or other proteins-dependent cell death, ROS, and callose deposition, promoted the expression of marker genes of SA or JA signaling pathway and *NbRbohb*, contributing to plants defense against pathogens. PlPeL1/PlPeL1-like enhanced pathogen virulence via their pectate lyase activity, which involves the degradation of plant cell wall components. Meanwhile, PlPeL1/PlPeL1-like were targeted by LcPIP1, potentially amplifying LcPIP1-induced plant cell death.

## Discussion

Plant cell walls serve as a crucial battleground in plant-pathogen interactions. Pathogens release a multitude of CWDEs, such as PeLs, to breach the cell wall barrier. Meanwhile, plants produce receptors and various CWDE inhibitors to trigger plant immunity and protect against damage[10]. However, the mechanisms by which plants respond to PeLs to mount defense responses remains broadly understudied.

In this study, two PeLs proteins, PlPeL1 and PlPeL1-like, coordinately contributed to the virulence of *P. litchii*. A growing number of studies support the critical role of PeLs in pathogenesis of plant

pathogens[15,17]. However, many studies have shown that deleting or silencing one or more PeL-encoding genes does not significantly impact pathogenicity[16,47]. It appears that the loss of function in one PeL gene can be compensated for by other PeLs[48]. In this study, individual deletions of *PlPeL1* or *PlPeL1-like* did not affect the pathogenicity of *P. litchii*, but deletion of both genes drastically reduced virulence (Fig. 1c). We deduced that the up-regulation of *PlPeL1* or *PlPeL1-like* in single-gene knockout mutants compensates for virulence defects (Supplementary Fig. 4). The structural arrangement of *PlPeL1* and *PlPeL1-like* as a bidirectional gene pair, head-to-head genes, implies the possibility of sharing regulatory elements, such as transcription factors, promoters, and enhancers[49]. Bidirectional promoters regulate the coordinated expression of gene pairs[50]. *PlPeL1* and *PlPeL1-like* exhibited similar expression patterns, implying a correlation in their expression. However, they displayed opposite transcription between 6-12 hpi (Supplementary Fig. 1e), which is consistent with the idea that there might be other mechanisms involved in the transcription regulation of bidirectional promoters[51].

We found that PlPeL1/PlPeL1-like interacted with LcPIP1 (Fig. 2), independent of their key enzymatic activity sites (Supplementary Fig. 6). LcPIP1 induced cell death in *N. benthamiana* leaves, but without any functionally known domain. PIP1s contributed to the defense of *N. benthamiana* and *A. thaliana* against *Ph. capsici*. As a positive immune regulator, PIP1 interacted with SERK3, which is a central membrane-localized receptor-like kinase (RLK) and plays a crucial role in countering pathogen invasion. PIP1s activated defense pathways that attenuated PlPeL1/PlPeL1-like induced *N. benthamiana*'s susceptibility to *Ph. capsici*, and PlPeL1$^{M1}$/PlPeL1-like$^{M1}$ could enhance LcPIP1-induced cell death (Figs. 3 and 4e). Notably, PlPeL1/PlPeL1-like relied on enzymatic activity to facilitate pathogen invasion, which may involve cell wall degradation. Consequently, the mere expression of LcPIP1/NbPIP1 may not consistently suffice to fully counteract the virulence potential of PlPeL1/PlPeL1-like under natural conditions, resulting in an incomplete elimination of the virulence impact of PlPeL1/PlPeL1-like in *P. litchii*. Consequently, the pathogenicity of ΔΔ*plpel1*/*plpel1-like* mutants was reduced. Additionally, it is possible that other effectors secreted by *P. litchii* inhibited the LcPIP1-induced response. A similar relationship exists between other pectinases and their interacting plant resistance proteins. For instance, plant polygalacturonase-inhibiting (PGIP) was inactivated by *Sclerotinia sclerotiorum* PGIP-inactivating Effector 1 (SsPINE1)[52]. Multiple CWDEs are essential virulence factors for pathogens, and also induce plant immune responses[42,53]. These findings emphasize the intricate dynamics and competitive interactions between plants and pathogens.

In this study, we observed a significant up-regulation of *LcPIP1*, *AtPIP1*, and *NbPIP1* when plants were exposed to biotic stress conditions, including infections by *P. litchii*, *Ph. capsici*, or *A. tumefaciens* (Figs. 4a, 8e, and Supplementary Fig. 7c). These results provide evidence for the involvement of PIP1s in plant responses to biotic challenges. Notably, the expression of either LcPIP1 or NbPIP1 significantly enhanced *N. benthamiana*'s resistance to *Ph. capsici*, but the *NbPIP1*-silenced plant did not exhibit significant susceptibility to *Ph. capsici*

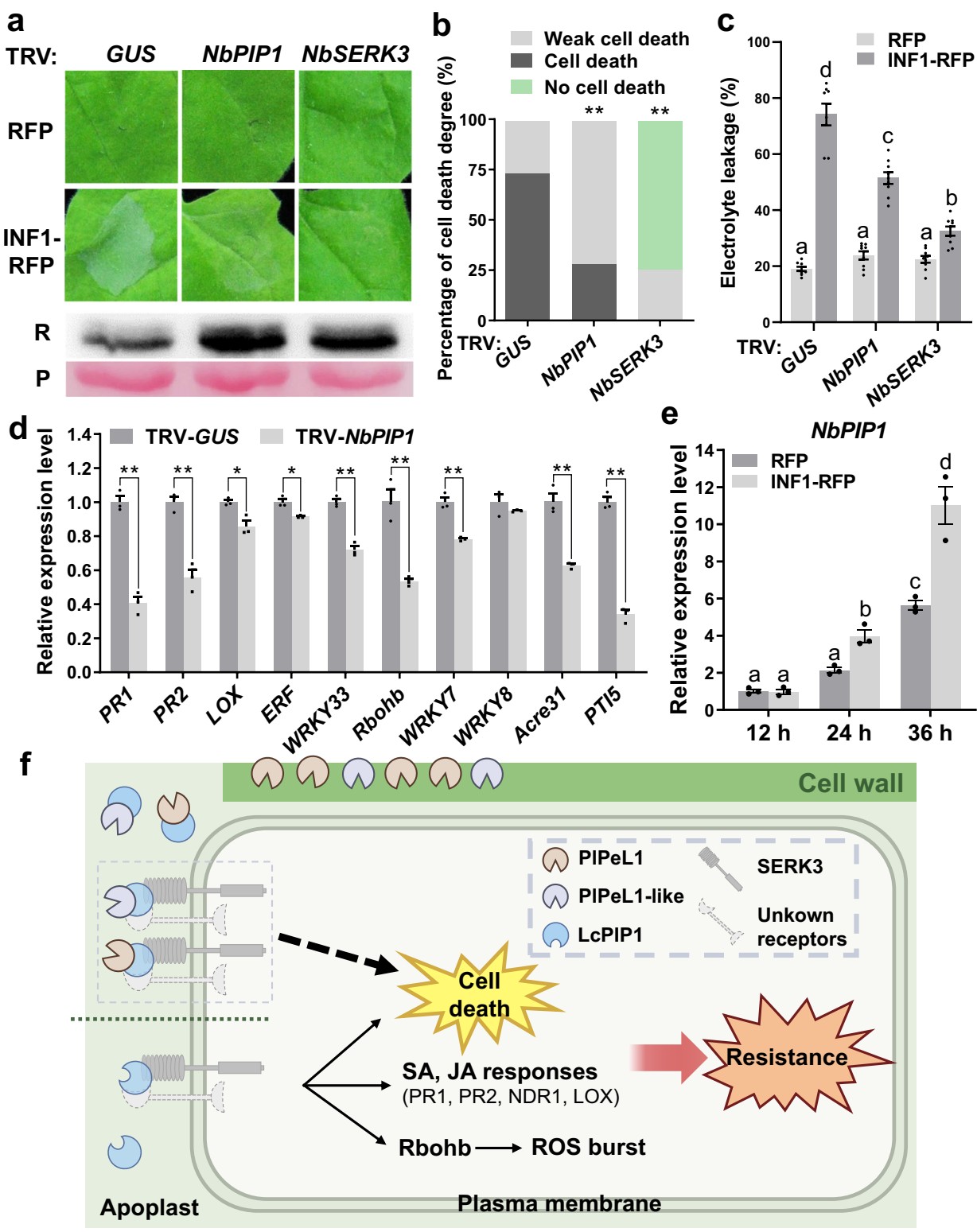

(Fig. 3). In addition, the *atpip1* mutants displayed enhanced susceptibility to *Ph. capsici* infection. One possible explanation is that silencing of *NbPIP1* using VIGS led to reduced expression, not complete elimination (Fig. 3d). With the complex network of plant defense responses in mind, another possibility is that the effects of *NbPIP1* silencing may be partially compensated by other proteins within the intricate defense network of the plant. Additionally, the absence of homologous proteins of PlPeL1 and PlPeL1-like in *Ph. capsici* could potentially contribute to the observed lack of a significant difference in resistance to

*Ph. capsici* between *NbPIP1*-silenced plants and the control. Furthermore, silencing of *NbPIP1* in *N. benthamiana* resulted in an increased susceptibility to *Ph. capsici* induction by PlPeL1, while no significant impact was observed with PlPeL1-like (Fig. 3j, k). Notably, LcPIP1 had a stronger effect in dampening susceptibility induced by PlPeL1 compared to its effect on PlPeL1-like (Fig. 3c). *PlPeL1* showed a greater than 20-fold increase in relative expression compared to *PlPeL1-like* at 24 hpi (Supplementary Fig. 1e), suggesting that the plant may require stronger activity to inhibit PlPeL1-mediated virulence. Nonetheless,

**Fig. 8 | Silencing of *NbPIP1* in *N. benthamiana* compromises INF1-triggered cell death. a** VIGS of *NbPIP1* delayed INF1-mediated HR development. INF1-RFP expressed in *NbPIP1*, *NbSERK3* or *GUS* -silenced plant leaves and photographs were taken at 2 dpa. Transient expression of INF1-RFP was confirmed by western blot, with anti-RFP antibody labeled "R" and Ponceau S labeled "P". **b** Quantification of INF1-mediated cell death in silenced plants. The degree of cell death was divided into three levels: weak cell death, cell death, and no cell death. Asterisks represent significant differences (**P < 0.01) based on Wilcoxon rank-sum test. **c** INF1-RFP and RFP (as a vector control for INF1-RFP) were expressed in *NbPIP1*-, *NbSERK3*- or *GUS*-silenced plants. Quantification of cell death by measuring electrolyte leakage at 2 dpa. Electrolyte leakage from infiltrated leaf discs was measured as a percentage of leakage from boiled discs. Data are shown as the mean ± SE (*n* = 8–9 biologically independent samples). Different letters on the graph represent significant differences among samples (One-way ANOVA; *P* < 0.05). **d** Relative expression of ten defense-related and PTI marker genes in TRV-*GUS* and TRV-*NbPIP1* leaves expressing INF1 at 2 dpa. The expression level of each gene in TRV-*GUS* leaves was set at 1. *NbEF1α* was used as the endogenous control. Data are shown as the mean ± SE of three replicates. Asterisks represent significant differences (**P* < 0.01, *P* < 0.05) based on Two-tailed Student's *t*-test. **e** qRT-PCR analyzed the expression levels of *NbPIP1* gene in *N. benthamiana* leaves expressing INF1-RFP or RFP at 12, 24, and 36 hpa. *NbEF1α* was used as the endogenous control. Data are shown as the mean ± SE of three replicates. Different letters on the graph represent significant differences among samples (One-way ANOVA; *P* < 0.05). All experiments were repeated three times with similar results. Source data are provided as a Source Data file. **f** Schematic model for the potential mechanism of PlPeL1/PlPeL1-like and LcPIP1 in plant immunity.

these underlying mechanisms remain elusive and necessitate further comprehensive investigation.

The receptor-like kinase SERK3 forms complexes with PAMP receptors and is the central regulator of innate immunity in plants[34]. SERK3 is involved in the regulation of various types of PCD[54]. SERK3 may function as a co-receptor or signaling regulator for a variety of receptor kinases and PRRs. For example, SERK3 forms complexes with BRI1 to activate BR signaling[55], with FLS2 or EFR to regulate a PTI pathway[56], and interacts with BON1 and CNGC20 in modulation of cell death[57,58]. In this study, we found that SERK3s interacted with PIP1s and were required for LcPIP1-induced cell death (Figs. 6 and 7). We speculate that LcPIP1 interacts with NbSERK3 to achieve signal transduction and activate the downstream immune response to trigger cell death in *N. benthamiana*. However, whether litchi can trigger immunity mediated by LcPIP1 and LcSERK3 by recognizing extracellular pathogen PeLs remains to be demonstrated. Moreover, LcPIP1-inducing cell death and plant immunity did not disappear completely in *NbSERK3*-silenced plants or *BAK1*-knockout lines (Fig. 6 and Supplementary Fig. 12). We speculate that LcPIP1-inducing cell death also depended on other plant receptors or signaling pathways. The interaction of LcPIP1[M3], the "VDMASG" motif-mutant, with LcSERK3 also provides evidence for this speculation (Supplementary Fig. 14). More research into the mechanism of LcPIP1-induced cell death is required.

HR triggered by INF1 has been shown to require the respiratory burst oxidase NbRbohb, MAPK kinase[59], and up-regulation of a large amount of genes transcription, such as PR (pathogenesis-related) protein genes, ET or JA-related genes[60]. NbSERK3 regulates the immune responses triggered by INF1[46]. In this study, silencing of *NbPIP1* also delayed elicitin INF1-induced HR (Fig. 8a–c). Expression of *NbPIP1* was induced by INF1 (Fig. 8e). We speculate that NbPIP1, interacting with NbSERK3, is a component of the protein complex responsible for INF1 perception and signal transduction. Notably, INF1-induced transcription levels of *WRKY8* showed no difference between *NbPIP1*- and *GUS*- silenced plants (Fig. 8d). WRKY7, WRKY8, WRKY9, and WRKY11 have all been identified as crucial components in INF1-mediated PTI ROS bursts[61]. Moreover, these WRKY proteins, including WRKY8, are known to redundantly contribute to the transactivation of RBOHB upon INF1 treatment[61,62]. In the *NbPIP1*-silenced plants, there was an observed delay in INF1-induced cell death, rather than complete inhibition. These findings could potentially explain the absence of transcriptional differences in *WRKY8* between *NbPIP1* and *GUS*-silenced plants. Further investigations are warranted to unravel the precise regulatory mechanisms governing *WRKY8* expression and its potential interaction with NbPIP1 in the context of the immune response.

In this study, *NbPR1*, *NbPR2*, and *NbNDR1* were significantly up-regulated in the *LcPIP1*-expressing leaves which are the marker genes of SA signaling pathway[31]. Moreover, NbEDS1 and NPR1 are two key components of the SA pathway[31,63]. Salicylic acid (SA) plays an essential role in hemi-biotrophic pathogen resistance and immune signaling via PRR[64,65]. Besides that, previous research has shown that high levels of SA facilitate programmed cell death[66]. We investigated whether the above SA-related genes function in LcPIP1-induced cell death. However, cell death induced by LcPIP1 was not significantly affected in silenced leaves of *NbEDS1*, *NbNDR1*, and *NbNPR1* (Supplementary Fig. 12). We hypothesize that the tested genes were not involved in LcPIP1-induced cell death.

In summary, our study unveils the dual roles of PlPeL1/PlPeL1-like in plant-pathogen interactions. PlPeL1/PlPeL1-like enhance pathogen virulence through pectate lyase activity while also being targeted by LcPIP1. LcPIP1/NbPIP1's role in attenuating PlPeL1/PlPeL1-like induced *N. benthamiana*'s susceptibility to *Ph. capsici* may have resulted from immune activation following their interactions. PIP1 acts as a positive regulator of plant resistance, inducing cell death and up-regulating resistance-related genes in *N. benthamiana*, likely through direct interaction with an established co-receptor NbSERK3. The interactions of PlPeL1/PlPeL1-like and PIP1s in *N. benthamiana* may induce a stronger immune response. To our knowledge, this is the first report on plants directly targeting plant pathogen PeLs to trigger immune response via a positive immune regulator, providing mechanistic insights into plant-oomycete interaction.

## Methods
### Plant and microbe cultivation
Litchi trees were cultivated in an orchard located at the South China Agricultural University in Guangzhou, China. *Nicotiana benthamiana* used in this study included wild-type (WT) and *BAK1*-knockout lines[43,44]. *N. benthamiana* was grown in the greenhouse at 16 h light per day of 21–25 °C. The Arabidopsis (*Arabidopsis thaliana*) Col-0 and knockout mutants were cultivated at 10–13 h light per day of 21–23 °C. *Peronophythora litchii* wild-type strain SHS21[21], Δ*plpel* mutants, and *Phytophthora capsici* strain LT263 were cultured on carrot juice agar (CJA) medium at 25 °C in the dark. For *P. litchii* sporangia production, five 9 mm diameter mycelial plugs were flushed with 2 mL sterilized water. The subsequent suspension was filtered using a 100 μm mesh filter and incubated at 16 °C for 1 h to induce zoospore release[67]. Plasmids were propagated in *Escherichia coli* strain JM109, which was cultured at 37 °C. *Agrobacterium tumefaciens* strain GV3101 was cultivated at 28 °C for agroinfiltration into *N. benthamiana*. *Saccharomyces cerevisiae* strain AH109 was used for yeast two-hybrid assay, culturing at 30 °C.

### Bioinformatics analysis
The genome sequence and gene annotations of *P. litchii* were obtained from NCBI (BioProject ID: PRJNA290406). The genome sequence of litchi was obtained from Sapinaceae Genomic (http://www.sapindaceae.com/)[68]. Protein sequences were submitted to the web-based tool Simple Modular Architecture Research Tool (SMART, http://smart.embl-heidelberg.de/) to identify conserved functional domains. Protein signal peptides (SP) were predicted using SignalP-4.1 or SignalP-5.0 (http://www.cbs.dtu.dk/services/). The alignment of

amino acid sequences was performed using BioEdit software. The phylogenetic dendrograms were constructed with the Neighbor-Joining algorithm (with the setting of 5000 bootstrap replications), by the MEGA 11.0 (http://megasoftware.net).

### PlPeL1 and PlPeL1-like genes expression patterns analysis

Total RNAs from different development stages of *P. litchii*, including mycelia and zoospores, and samples from 3, 6, 12, and 24 h post-inoculation with zoospores on leaves, were extracted using the All-In-One DNA/RNA Mini-preps Kit (Bio Basic, Shanghai, China). Reverse transcription was performed using SYBR® Premix Ex Taq™ II (TaKaRa, Japan). Expression profiles of *PlPeL1 and PlPeL1-like* genes were detected using quantitative reverse-transcription polymerase chain reaction (qRT-PCR) on qTOWER³ Real-Time PCR thermal cyclers (Analytik Jena, Germany) with SYBR® Premix Ex Taq™ II (TaKaRa, Japan). The *P. litchii* actin gene (*PlActin*)[69] was used as an endogenous control, and the relative fold change was calculated using the $2^{-\Delta\Delta CT}$ method. Primers used for these analyses were listed in Supplementary Data 4.

### Peronophythora litchii transformation and virulence assays in litchi leaves

The vectors pYF2.3G-Ribo-sgRNA and pBluescript II KS were utilized for the knockout of *PlPeL1* and *PlPeL1-like* using the CRISPR/Cas9 genome-editing system. Four sgRNAs were designed, with each pair targeting both *PlPeL1* and *PlPeL1-like*. Subsequently, the constructs and pYF2-PsNLS-hSpCas9 were introduced into wild-type SHS3 protoplasts of *P. litchii* using the PEG-mediated transformation method[38,67]. The transformed protoplasts were regenerated overnight, and the recovered mycelia were selected on CJA medium supplemented with 40 μg/mL G418 at 25 °C in the dark. Transformants were verified by genomic PCR and sequencing. Primers and constructs used for these analyses were listed in Supplementary Data 4 and Data 5. Pathogenicity assays were performed by applying suspensions containing 200 zoospore each of *P. litchii* WT and knockout-mutants on litchi leaves at 25 °C in the dark. Lesion diameters were recorded and photographed 48 h after inoculation. The significance of differences was analyzed with one-way ANOVA test and three independent replicates were set up, with at least six leaves in each replicate.

### N. benthamiana RNA extraction and qRT-PCR

*N. benthamiana* leaf samples (n = 3) were ground in liquid nitrogen. The methods for RNA extraction and reverse transcription are as described above. The translation elongation factor 1 alpha (*EF1α*) was used as the endogenous control and the primers are listed in Supplementary Data 4.

### Agrobacterium tumefaciens-mediated transient expression in N. benthamiana

*PlPeL1 and PlPeL1-like* genes were amplified from *P. litchii* cDNA and inserted into the pBin plasmid with an RFP or HA tag using ClonExpress MultiS One Step Cloning Kit (Vazyme, China). Primers and constructs used for these analyses were listed in Supplementary Data 4 and Data 5. Recombinant plasmids were transformed into the *A. tumefaciens* GV3101 by heat shock. Transformants were verified by PCR and cultured in Luria Bertani (LB) broth with rifampicin (25 μg/mL) and kanamycin (50 μg/mL) at 28 °C 200 rpm for 36 h. The *A. tumefaciens* cells were collected and resuspended three times with infiltration buffer (10 mM MgCl₂, 10 mM MES, pH 5.7, and 100 μM acetosyringone) to a final $OD_{600}$ of 0.4–0.6. The mixed cell suspensions were infiltrated into the leaves of *N. benthamiana* using blunt syringes. All the experiments have been repeated at least three times.

### Enzyme activity assays of pectate lyase

Pectate lyase activity was assayed with 0.25% (w/v) polygalacturonic acid (PGA; Sigma) as the substrate[40]. Target proteins were expressed

in *N. benthamiana* leaves using agroinfiltration. A total of 0.1 g of infiltrated leaf tissue was collected at 2 dpa, and subjected to protein extraction with 1 mL pectate lyase extraction buffer (Cominbio, China). The reaction mixture of untreated leaf tissue was used as a negative control. An aliquot of 100 μL extraction solution was added to 900 μL of reaction buffer (0.5% PGA, 50 mM glycine-NaOH, pH 9.5, and 1 mM CaCl₂) and the solutions were measured using spectrophotometer (Thermo Scientific, USA) at 235 nm. The sample was then kept at 42 °C for 30 min. PeL was assayed by measuring the increase in absorbance at 235 nm. One unit (U) of enzyme activity was defined to be the amount of enzyme required to release 1 μmol of oligogalacturonic acid from PGA per minute at 42 °C, pH 9.5. Enzyme units were calculated using the formula, Units/g (measured sample) = (Absorbance after the reaction - Absorbance before the reaction) / (4600 L/mol/cm × 1 cm) × 0.001 L × 109/(0.1 mL × 0.1 g/1 mL)/30 min - Units/g (negative control). The molar absorption coefficient of the oligogalacturonides was 4600 L/mol/cm.

### Generation of amino acid mutants of PlPeL1, PlPeL1-like, and LcPIP1

The generation of amino acid mutants for PlPeL1, PlPeL1-like, and LcPIP1 was carried out through a polymerase chain reaction (PCR) method. Primers were designed to encompass both the desired mutation site and its flanking sequences. Primers were listed in Supplementary Data 4. The PCR thermal cycling profile included an initial denaturation step at 95 °C for 5 min, followed by 34 cycles of denaturation at 95 °C for 30 seconds, annealing at 60 °C for 30 seconds, and extension at 72 °C for 30 seconds. A final extension step was performed at 72 °C for 5 min. After purifying the PCR product, the amplified DNA fragments were inserted into the enzyme-digested vector through homologous recombination, using (ClonExpress Ultra One Step Cloning Kit (Vazyme, China). The resulting recombinant DNA constructs, containing the desired amino acid mutation sequences, were then transformed into competent *E. coli* cells. Positive clones were selected and verified through DNA sequencing.

### Phytophthora capsici inoculation assay

Target proteins were expressed in *N. benthamiana* leaves via agroinfiltration as described above. The infiltrated regions were inoculated with mycelia plugs (diameter = 5 mm) of *Ph. capsici* 24 h post infiltration in the dark. Lesion diameters were recorded and photographed under ultraviolet (UV) light at 48 h after inoculation. After analysis of variance with three replicates composed of 10 leaves each, one-way ANOVA was used to identified significantly different treatments at *P* = 0.05.

The *A. thaliana* leaves were inoculated with zoospores of *Ph. capsici*, following the procedure outlined by Li et al[70]. Specifically, to prepare *Ph. capsici* zoospores, mycelium was incubated in 10% (v/v) V8 broth at 25 °C for 2–3 days, followed by three washes in autoclaved mineral water at room temperature. After 2 days of continuous light incubation, numerous sporangia were formed. To induce zoospore release, the cultures were subjected to a temperature shift, first at 4 °C for 30 min followed by incubation at 25 °C for 30 min. For zoospore inoculation, *A. thaliana* leaves were cut and incubated with 150 *Ph. capsici* zoospores at 25 °C in the dark.

### Phytophthora capsici biomass detection

For biomass analysis, the infected *N. benthamiana* or *A. thaliana* leaves were ground in liquid nitrogen. Total DNA was extracted from the same areas around *Ph. capsici*-infected sites and used for quantitative PCR (qPCR) to quantify the ratio of host-to-pathogen biomass, using primers specific for plant *NbEF1α* or *AtUBC9*, and *Ph. capsici Actin* genes (Supplementary Data 4).

## Generation of *atpip1* mutants

To knockout *AtPIP1* (*AT5G58375*), the CRISPR/Cas9-mediated genome-editing method was used[41] with sgRNA (sg-*AtPIP1*: CAT-ATTCCTTTCAGCGACTC). The genotypes of the transgenic plants were analyzed by PCR and Sanger sequencing using DNA extracted from transgenic plants as template, and primers used for these analyses are listed in Supplementary Data 4.

## Yeast two-hybrid assay

A cDNA library derived from litchi leaves inoculated with *P. litchii* zoospore suspensions was constructed in the yeast vector pGADT7 (AD) (Clontech) for prey protein screening. *PlPeL1* gene was cloned into the yeast vector pGBKT7 (BD) (Clontech) as bait. Recombinant BD-PlPeL1 plasmid was co-transformed into yeast cells (strain AH109) with AD-library and were selected using SD/-Trp/-Leu/-His or SD/-Trp/-Leu/-His/-Ade medium. The gene sequences of candidate targets were verified by individual colony PCR and sequencing. *LcPIP1* was cloned into the vector pGADT7 and co-transformed into AH109 with pGBKT7-*PlPeL1* and then tested on SD/-Trp/-Leu medium and SD/-Trp/-Leu/-His/-Ade medium with X-α-gal. BD-53 + AD-T and BD-Lam+AD-T were used as the positive and negative controls[71], respectively.

## Co-immunoprecipitation and western blot analysis

Proteins were extracted from the leaves of *N. benthamiana* after agroinfiltration using protein extraction buffer (no. P0013; Beyotime, China) supplemented with 0.1% (v/v) protease inhibitor cocktail (Sigma-Aldrich). The mixture was centrifuged at 4 °C 13000 g for 10 min, and the supernatant was transferred to a new tube. For Co-IP assays, 15 μL RFP-Trap-M beads (ChromoTek, China) were added to the supernatant and incubated at 4 °C for 1 h. After incubation, the beads were washed five times with protein extraction buffer. Subsequently, the beads were boiled with 40 μL of PBS and 10 μL of SDS-PAGE sample loading buffer for 10 min. The eluted proteins were then subjected to SDS-PAGE gel separation, followed by protein analysis using western blot.

## Western blot analyses

Total proteins were separated using SDS-PAGE gels, followed by transfer to polyvinylidene fluoride (PVDF) membranes (Bio-Rad). The PVDF membranes were then blocked with 5% non-fat milk in PBST (phosphate-buffered saline + 0.1% Tween 20). Mouse anti-RFP-tagmonoclonal antibodies (ChromoTek, China), anti-Flag-tag, or anti-HA-tag monoclonal antibodies (Abmart, China) were used to detect the corresponding fusion proteins at a 1:5,000 dilution. After three washes with PBST, a goat anti-mouse antibody (1:10,000) was applied (Abmart, China). Finally, proteins were visualized using the Efficient Chemiluminescence kit, and photographs were taken with an imaging system (Bio-Rad ChemiDoc XRS + , USA).

## In vitro GST pull-down

The plasmids pGEX-6P-1, pGEX-6P-1-*PlPeL1*, pGEX-6P-1-*PlPeL1-like*, pET32a-*LcPIP1*, and pET32a-*NbPIP1* were expressed in *E. coli* strain BL21 to produce GST, GST-tagged PlPeL1/PlPeL1-like and His-tagged LcPIP1/NbPIP1 proteins at 18 °C and 180 rpm, in 0.2 mM isopropyl-β-D-thiogalactopyranoside (IPTG) for 10 h. The bacteria lysate containing GST-PlPeL1 or GST-PlPeL1-like was incubated with 40 μL GST magnetic beads (Thermo Fisher Scientific) for 1 h at 4 °C. After washing the beads five times with PBS, they were incubated with His-tagged proteins for 2 h at 4 °C. The beads were washed five times again, and the presence of His-LcPIP1 and His-NbPIP1 proteins was detected by western blot using anti-His antibody.

## Collection of plant apoplastic fluid

The *N. benthamiana* apoplastic fluid was collected 2 days after agroinfiltration using Phosphate Buffered Saline (PBS) buffer (50 mM NaPO$_4$, 150 mM NaCl, pH = 7.0), following Dong et al.[72]. The samples were placed into a vacuum desiccato (Eppendorf, Germany), and a pressure of 60 hPa was applied for 5 min using a pump. After removing PBS buffer from the leaf surface using paper towels, the leaves were centrifuged at 1000 g for 6 min at 4 °C, and the apoplastic fluid was pooled into a tube. Subsequently, the *N. benthamiana* apoplastic fluids were filter-sterilized using 0.45 μm filters and either used immediately or stored at −80 °C. Proteins in the apoplastic fluid were determined by western blot.

## Yeast signal trap assay

The function of the signal peptides (SPs) of PlPeL1/PlPeL1-like and LcPIP1 were tested in *Saccharomyces cerevisiae* YTK12 strain, using the pSUC2 vector, following the procedure outlined by Yin et al.[37]. These SPs were integrated into the pSUC2 vector to create an in-frame fusion with invertase. The resulting vector were introduced into the yeast strain YTK12 and cultivated on CMD-W plates. Invertase secretion was detected by examining yeast cultures grown in YPAD liquid medium using the transparent triphenyltetrazolium chloride (TTC) assay. YTK12 strains were transformed with pSUC2-SPAvr1b or pSUC2 as positive and negative controls, respectively.

## ROS and callose staining

For ROS staining, *N. benthamiana* leaves were collected 36 h after Agrobacterium infiltration and stained with 1 mg/mL 3,3′-diamino-benzidine (DAB) for 8 h in the dark and then decolored with ethanol. For the callose deposition assay, leaves were decolored with ethanol and stained for 1–2 h in 150 mM K$_2$HPO$_4$ buffer with 0.01% aniline blue. A microscope (Olympus, Japan) was used to take photographs. ImageJ software was used to assess ROS accumulation and callose deposition. At least three leaves were tested in each independent experiment. The experiments were repeated at least three times.

## Electrolyte leakage

Leaves of *N. benthamiana* were cut into discs of 6 mm in diameter, which were floated on distilled water for 3 h at room temperature. The electrical conductivity (EC1) of the solution was measured with a meter (DDS-307A; Rex Shanghai). The sample was then boiled for 30 min and cooled to room temperature. The second electrical conductivity measurement (EC2) of the solution was taken. Electrolyte leakage was calculated as follows: electrolyte leakage (%) = EC1/EC2 × 100[73].

## Accession numbers

Gene nucleotide and protein sequences used in this study were obtained from NCBI (https://www.ncbi.nlm.nih. gov), TAIR (https://www.arabidopsis.org/), Sapinaceae Genomic (http://www.sapindaceae.com/)[68], UniProt (https://www.uniprot.org/), or Sol Genomics Network (http://solgenomics.net/) databases. Accession numbers are as follows: PlPeL1, OR241278; PlPeL1-like, OR241279; LcPIP1, LITCHI031111; LcPIP1PP, LITCHI003344; LcSERK3, LITCHI011582; NbSERK3A, Niben101Scf11779g01002.1; NbSERK3B, Niben101Scf00279g02022.1; NbBAK1a, Niben101Scf02128g00022.1; NbBAK1b, Niben101Scf02513g11004.1; NbPIP1, Niben101Scf11235g01010; AtPIP1, AT5G58375; AtBAK1, AT2G23950.

## Reporting summary

Further information on research design is available in the Nature Portfolio Reporting Summary linked to this article.

# Data availability

The mRNA sequences for PlPeL1 and PlPeL1-like have been submitted to NCBI and assigned accession numbers OR241278 and OR241279, respectively. Data supporting the findings of this work are available within the paper and its Supplementary Information files. Source data are provided in this paper.

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

## Acknowledgements

This work was supported by the China Agriculture Research System (CARS-32), the National Natural Science Foundation of China (31701771), and the Natural Science Foundation of Guangdong Province, China (2022A1515010458). We thank Professor Brett Tyler (Oregon State University, United States) for offering plasmids. We appreciate Professors Yuanchao Wang and Yan Wang (Nanjing Agricultural University, China) for offering *BAK1*-knockout *N. benthamiana* seeds. We are very grateful to Professor Tom Hsiang (University of Guelph, Canada) for the English revisions of the manuscript.

## Author contributions

Z.D.J., G.H.K., and W.L. designed the research; W.L., P.L., J.J.S., Z.Y.H., W.Z.Z., M.H.L., and P.G.X. performed the research; W.L., P.L., and Y.Z.D. analyzed the data; W.L., G.H.K., and J.J.S. wrote the paper; W.L., P.L., Y.Z.D., X.X.L., G.H.K., and Z.D.J. discussed and revised on the manuscript; All authors read and approved the submitted version.

## Competing interests

The authors declare no competing interests.
