## [Peer Review File · Nature Communications]

A plant cell death-inducing protein from litchi interacts with
Peronophythora litchii pectate lyase and enhances plant
resistanceReviewer #1 (Remarks to the Author):

The manuscript by Li et al describes the identification of two putative pectate lyases from the oomycete *Peronosphythora litchi* (PIPELs) whose function is required for virulence, as well as the identification of LcPIP1, a protein from litchi that interacts with both PIPELs and that induces cell death when overexpressed in *N. benthamiana*. The authors go on to show that orthologues of LcPIP1 also induce cell death in *N. benthamiana*, and report that LcPIP1 interacts with BAK1/SERK3, and that the latter is required for LcPIP1-mediated cell death induction. The authors propose a model where LcPIP1 will be a positive regulator of immune responses that would interact with SERK3. The manuscript contains a very important amount of work. Some experiments, like the characterization of *P. litchii* mutants are of high standard, clear and support the claims, whilst other experiments raise some questions. The identification of LcPIP1 and of its orthologues in other plant species is noteworthy and suggests that the findings can be of general interest in the field of plant pathogen interactions. However, given the importance of the claims, other methodologies and/or experiments might be necessary to sustain them. For example, instead of gene silencing experiments, using CRISPR/Cas9 editing of SERK3/BAK1 in *N. benthamiana* and/or *A. thaliana* mutants will shed light on the role of SERK3 in the interaction. The same could be done for NbPIP1 and AtPIP1.

One of my concerns with the manuscript is that the methods are very limited, lacking in detail and even absent for several experiments. This, other than not allowing other researchers reproducing the work, makes difficult understanding some parts of the manuscript and complicates the task of assessing if the results presented justify the claims. Two examples of absences and their consequences follow:

- The material and methods section for the qRT-PCRs for plant genes is absent. In the qRT-PCR analysis presented in Figures 3a and 4, the values for CK (Fig 3a) and RFP (Fig 4) appear to be the same at all time points. This is surprising, particularly for Fig 4, because agroinfiltration should give some level of induction of PTI-related genes. In absence of the methods, one possible explanation would be that the authors calculated the relative expression of the genes for each time point using the control for each time point as a reference, instead of using the earliest time point for controls for all samples. If this was the case, this would be misleading concerning the time-course of gene expression. This point needs to be clarified and corrected if required.
- The material and methods section for most of the cloning is absent. In the text the authors describe creating PIPEL-M2s mutants, which imply mutating 7 Asp to Ala. There is no description in the methods of how this was achieved (one could guess how this was done from the primer list provided in supplementary data, but it is not specified in the methods). The same applies to most of the cloning done.

Another main concern has to do with the versions of the protein carrying or not a signal peptide (SP). In P8 and Figure 1 the authors claim that the PIPEL1 mutant lacking the SP (M1) lost its virulence function. This would make sense because PIPEL1 is meant to function in the extracellular space, and absence of SP means that the protein will not be secreted. However, on Suppl Fig 6, M1 is barely detectable compared to the WT, so it is difficult to conclude that this mutant lost its virulence function. The same applies to LcPIP1 Δ SP: the fact that the protein is barely detectable doesn't allow to draw a conclusion about the role of the SP in cell-death induction (Figure 3). This leads to the point that the authors didn't test the functionality of the PIPEL1 SP. This is important because, based on the model proposed for the authors in Figure 8, PIPELs and LcPIP1 function in the extracellular space, but in reality it is not clear if PIPELs is being secreted when transiently expressed in *N. benthamiana*. For LcPIP1 the authors show that it is partially secreted in yeast. Concerning its expression in *N. benthamiana*, looking at Figure 3d, the detected LcPIP1 protein appears bigger than LcPIP1 Δ SP, suggesting that the signal peptide is still present in the protein. Because SP-mediated secretion involves cleavage of the SP, this would mean that most of the detected protein is still intracellular, which will be difficult to reconcile with the protein interacting with the extracellular part of SERK3.

In figure 3d it is also noteworthy that the protein carrying the PR1-SP appears smaller than LcPIP, when it should be at least identical (if not bigger, based on the cartoon in Fig 3c). This could be explained if the PR1-SP was processed, but then the size should be the same as LcPIP1 Δ SP, which is not. All this needs to be clarified.

Related to this, in several CO-IP experiments, the IP for the target protein shows several bands, some of them present at the same intensity as the target protein, which are not present in controls (see Figures 7 and Suppl 7). What could these proteins be and how do the authors rule out that the interaction does not take place with one of these proteins instead of with the target protein?

Finally, in most cases the authors present their results as histograms with the means \pm SE of independent biological replicates. I suggest adding the data points to the histograms in order to have a better view of the dispersion of the data points, and identifying the different repeats (alternatively use boxplots or other presentation that helps visualizing dispersion). Numbers of datapoints should be indicated for all experiments. I assume that the authors confirmed that their dataset fulfils the requirements for the statistical test used.

Other suggestions

Abbreviate differentially Phytophthora and Peronophythora

Overall, since the authors have not demonstrated pectate lyase activity, it might be more appropriate using the terms putative or candidate pectate lyases.

Overall, the authors use lesion diameter to account for pathogen infection, but lesions are not always circular, in particular for experiments on litchi. Using lesion area would be more accurate.

Overall, the figure legends are limited. This is particularly true for the legends of the Supplementary Figures.

P7, l 118-122. The claim that up-regulation of either PIPeL1 and PIPeL1-like upon knocking out the other supports functional redundancy should be played down. In Suppl Fig 1 the authors show that both genes are separated by approx. 1100 bp. This proximity in the promoters for both genes, even if they are in opposite strands, could lead to a competition for expression in the WT, and thus to an increase of the expression of one of the genes when the other is knocked down.

P7, l 130-131 and Suppl Fig 5. In order to highlight the conservation of the Asp residues, it could be more interesting to present the alignment of PIPeLs with other PeLs from other species. Related to this, the sentence in l 130 needs to be developed.

Define positive and negative controls in Figure 2a

P11, l 208. The conserved motif VDMASG could be extended to VDMASGK/RYLW

P11, l 209. It is not clear to me what is the contribution to the manuscript of the phylogenetic tree presented in Figure 5a.

P11, l 209. Provide accession numbers for all proteins in Suppl Figure 9.

P15, l 297. NbPIP1 is upregulated upon INF-1 mediated cell death (Figure 8d) as well as upon PIPeL1 expression (Suppl Figure 11). INF1 induces defense responses and PIPeLs suppresses defense responses. How does it fit with the model of NbPIP1 function?

As a non-native English speaker, I understand how frustrating this can be, but the manuscript would benefit from proof-reading by a native English speaker.

Reviewer #2 (Remarks to the Author):

This MS describes the identification and characterisation of two secreted proteins (PIPeL1 and PIPeL-

like proteins) from *Peronophythora litchii* and their target.

In short, the authors show that:

- 1- deletion of both genes in *P. litchii* results in a loss of virulence
- 2- Said PI proteins interact with a small secreted protein (Pip1) in plants
- 3- LcPIP mediates cell death (orchestrated by a small and highly conserved motif)
- 4- Plant cell death resembles a PTI-type response that requires SERK3 and activate expression of known PTI responsive genes

Based on these observations, the authors conclude that LcPIP is a cell death inducing peptide that interacts with two PI effectors.

Overall, the work is presented nicely and the MS is of good written quality. Major comments are listed below:

- It is not clear to me what the main message of the work is and how it is important. LcPIP induces cell death when over-expressed in plants and its silencing compromises immunity to virulent pathogens. This is an interesting observation (bolstered by interesting interaction data). Does this reduced immunity phenotype extend to non-pathogenic microbes? Reduction of PTI gene expression hints at a more general impairment, but this is not explored. That's where I think exploration would lead to exciting discoveries.

- The authors show that two PI effectors bind Litchi (and Nb) PIP AND that disabling both effectors affect virulence. The significance of these observation is not clear. What is the consequence of PIP1 binding? Disabling of PIP1 function or PAMP perception? The authors will know that knocking out two effector coding genes could cause a virulence phenotype that has little to do with LcPIP1 signalling. Infection assays that feature knockout strains and PIP1 mutants would help address that.

It is possible that PIP1-mediated immunity is actively suppressed during infection and if so, understanding the mechanisms by which PI disrupts its signalling would be of great interest. As it stands, however, there is not enough data to support a credible model that informs on the action of PIP1 and (possibly) PI virulence factors. The proposed model reflect these knowledge gaps as it is fairly vague and throws up many questions.

Reviewer #3 (Remarks to the Author):

This manuscript describes the involvement of pectate lyases (PIPeL1 and PIPeL1-like) from *Peronophthora litchii* in pathogenicity and a novel plant cell death-inducing protein, LcPIP1, from litchi. First, the authors created single or double deletion mutants of PIPeL genes and demonstrated that the double deletion mutant impaired the pathogenicity of *P. litchii*. Next, the authors investigated the interaction of PIPeLs and PIP1s. They demonstrated that the signal peptides and the VDMASG motif of LcPIP1 were essential to induce PCD in *Nicotiana benthamiana*. Moreover, they showed that the plant immune response by LcPIP/NbPIP was triggered via SERK3. The data presented in this manuscript is very interesting and the methods used are reasonable. The manuscript, however, has several critical issues. I would recommend this manuscript for publication in *Nature Communications*, provided the authors reasonably address the following points.

Line 91:

Is the identity (99.78%) ranged from 1 to 460 amino acids?

These two pel genes can be named as PIPeL1 and PIPeL2 because the C-terminal regions of the

two proteins are quite different. That makes easier to distinguish them in the manuscript. The authors don't have to change the names but I strongly recommend.

Line 125:

Was the full length of PIPeL1-like protein expressed including the unique C-terminus?

Line 142:

Which means that the increase of the susceptibility was responsible for only the degradation of pectic substances (pectate)?

Do PIPeL1M2 and -likeM2 contain signal peptides?

Line 166:

Have you conducted the expression assay of LcPIP1 using litchi inoculated with a single or double deletion mutants of Pel genes?

Apparently, Pels up-regulate NbPIP1 in *N. benthamiana* from the results of Supplementary Fig. 11. Whether the expression of LcPIP1 is also regulated by Pels in litchi is intriguing.

Line 233:

Single (PIP) and co-expressing (PIP/Pel) leaves showed the same lesion size, which raises the following question: PIPeL1 and -like have no effect on the infection of *P. capsici*?

I just wonder how the complex of LcPIP1/PIPeL1 is involved in pathogenicity and plant resistance. I understand that LcPIP1 indeed induced PCD in *N. benthamiana* from your data. Do you expect that PIPeLs from *P. litchii* have function to inhibit LcPIP1 from binding to SERK3 through interacting with each other in addition to degradation of pectic substances? or LcPIP is secreted to inhibit the enzymatic activity of PIPeLs in addition to the induction of immune systems? I understand that they have not completely proven yet in your work, but I believe it is necessary to describe your prediction at least. From the title of your paper, readers will imagine the effect given by the interaction between Pel and PIP toward pathogenicity and the plant immune systems.

Line 251:

Do you mean that the susceptibility induced by PIPeL1 is different from that by PIPeL1-like in mechanisms? If NbPIP1 has higher ability to attenuate the susceptibility induced by PIPeL1, I just wonder why leaves expressing PIPeL1 or PIPeL1-like, expressing PIP1 as well, showed the same lesion size (Figure 1d-g and Figure 6ab).

If you know, please let me know. Is the homolog of INF1 from *P. capsici* secreted in inoculated *N. benthamiana*?

Line 295:

What do you think is the reason why the expression of WRKY8 was not influenced? Please state about it in Discussion.

Line 331:

Please add 'in *N. benthamiana*' at the end of the sentence.

Line 365:

As you mentioned, NbPIP1 may interact with INF1 to trigger the immune systems. Again, how about the interaction (binding) between Pel1 and PIP1? What does this interaction cause? Just to avoid the degradation of cell walls?

Line 379:

You mentioned that NbPIP1 attenuated the susceptibility induced by PIPeL1 in line 251 (Figure 6g,h). But here, you describe that NbPIP1-silencing did not affect the susceptibility quoting Figure 6g,h. It is confusing.

Line 386:

Can you say PIPeLs trigger the immune systems?

If so, leaves expressing PIPeLs should show resistance against *P. capsici* somehow.

Your sentence from lines 386 to 388 sounds like it is happening in litchi.

I suggest that you describe the roles of PelLs from the pathogen end and those of PIP1 from the

plant end, and on top of that, what the interaction of PeLs and PIP1 brings about.
Please distinguish clearly whether you are describing about *N. benthamiana* or litchi.

REVIEWER COMMENTS

Reviewer #1 (Remarks to the Author):

The manuscript by Li *et al* describes the identification of two putative pectate lyases from the oomycete *Peronophythora litchi* (PIPeLs) whose function is required for virulence, as well as the identification of LcPIP1, a protein from litchi that interacts with both PIPeLs and that induces cell death when overexpressed in *N. benthamiana*. The authors go on to show that orthologues of LcPIP1 also induce cell death in *N. benthamiana*, and report that LcPIP1 interacts with BAK1/SERK3, and that the latter is required for LcPIP1-mediated cell death induction. The authors propose a model where LcPIP1 will be a positive regulator of immune responses that would interact with SERK3. The manuscript contains a very important amount of work. Some experiments, like the characterization of *P. litchii* mutants are of high standard, clear and support the claims, whilst other experiments raise some questions. The identification of LcPIP1 and of its orthologues in other plant species is noteworthy and suggests that the findings can be of general interest in the field of plant pathogen interactions. However, given the importance of the claims, other methodologies and/or experiments might be necessary to sustain them. For example, instead of gene silencing experiments, using CRISPR/Cas9 editing of SERK3/BAK1 in *N. benthamiana* and/or *A. thaliana* mutants will shed light on the role of SERK3 in the interaction. The same could be done for NbPIP1 and AtPIP1.

Re: We would like to express our gratitude to the reviewer for valuable time and thoughtful feedback, which has greatly contributed to the improvement of our manuscript. In the revised manuscript, we incorporated data from *Arabidopsis thaliana* and *Nicotiana benthamiana* mutants generated through CRISPR/Cas9 editing.

Our results showed that AtPIP1 also interacts with AtBAK1 (new Supplementary Fig. 13b) and loss of *AtPIP1* led to higher susceptibility to pathogens (new Fig. 3g-i).

Deletion of *NbBAK1* significantly reduced cell death and compromised the plant's resistance induced by LcPIP1 (new Fig. 6). These results made our results more solid and demonstrated that PIP1 and its homologs from different plants could positively contribute to plant immunity, which suggest that the findings can be of general interest in the field of plant pathogen interactions.

One of my concerns with the manuscript is that the methods are very limited, lacking in detail and even absent for several experiments. This, other than not allowing other researchers reproducing the work, makes difficult understanding some parts of the manuscript and complicates the task of assessing if the results presented justify the claims. Two examples of absences and their consequences follow:

- The material and methods section for the qRT-PCRs for plant genes is absent. In the qRT-PCR analysis presented in Figures 3a and 4, the values for CK (Fig 3a) and RFP

(Fig 4) appear to be the same at all time points. This is surprising, particularly for Fig 4, because agroinfiltration should give some level of induction of PTI-related genes. In absence of the methods, one possible explanation would be that the authors calculated the relative expression of the genes for each time point using the control for each time point as a reference, instead of using the earliest time point for controls for all samples. If this was the case, this would be misleading concerning the time-course of gene expression. This point needs to be clarified and corrected if required.

- The material and methods section for most of the cloning is absent. In the text the authors describe creating PIPeL-M2s mutants, which imply mutating 7 Asp to Ala. There is no description in the methods of how this was achieved (one could guess how this was done from the primer list provided in supplementary data, but it is not specified in the methods). The same applies to most of the cloning done.

Re: We thank the reviewer for the comments, and we have taken steps to address these concerns in our revised manuscript. Specifically, we have revised the manuscript to include a more detailed description of the experimental methods utilized in the revised Methods section. Furthermore, we have revised our figures (new Fig. 5) and relative sentences for curate expression (figure legend of Fig. 5). Moreover, we have included additional details on the generation of amino acid mutants for PIPeL1, PIPeL1-like, and LcPIP1 in the "Generation of amino acid mutants of PIPeL1, PIPeL1-like, and LcPIP1" section of the Methods.

Another main concern has to do with the versions of the protein carrying or not a signal peptide (SP). In P8 and Figure 1 the authors claim that the PIPeL1 mutant lacking the SP (M1) lost its virulence function. This would make sense because PIPeL1 is meant to function in the extracellular space, and absence of SP means that the protein will not be secreted. However, on Suppl Fig 6, M1 is barely detectable compared to the WT, so it is difficult to conclude that this mutant lost its virulence function. The same applies to LcPIP1 Δ SP: the fact that the protein is barely detectable doesn't allow to draw a conclusion about the role of the SP in cell-death induction (Figure 3).

Re: We expressed full length of PIPeL1/PIPeL1-like with a lower agrobacterium concentration ($OD_{600}=0.0005$), and the protein accumulation of PIPeL1/PIPeL1-like was still higher than that of M1 mutants (Author Response Figure 1, for review only). Indeed, the loss of virulence in PIPeL1 Δ SP /PIPeL1-like Δ SP is very likely due to the low protein accumulation. Therefore, we have removed the results of leaves expressing PIPeL1 Δ SP or PIPeL1-like Δ SP inoculated with *Ph. capsici*. Meanwhile, we have renamed the mutant with amino acid replacement (7 Asp to Ala) as M1.

Author Response Figure 1 (For review only) Immunoblot analysis of PIPeL1^{ΔSP} and PIPeL1-like^{ΔSP} expressed in *N. benthamiana* leaves.

Total proteins were extracted from *N. benthamiana* leaves at 36 hpa. Red asterisks indicated protein bands of the correct size. Ponceau S staining of Rubisco was used to indicate loading quantity of protein samples.

This leads to the point that the authors didn't test the functionality of the PIPeL1 SP. This is important because, based on the model proposed for the authors in Figure 8, PIPeLs and LcPIP1 function in the extracellular space, but in reality it is not clear if PIPeLs is being secreted when transiently expressed in *N. benthamiana*. For LcPIP1 the authors show that it is partially secreted in yeast. Concerning its expression in *N. benthamiana*, looking at Figure 3d, the detected LcPIP1 protein appears bigger than LcPIP1^{ΔSP}, suggesting that the signal peptide is still present in the protein. Because SP-mediated secretion involves cleavage of the SP, this would mean that most of the detected protein is still intracellular, which will be difficult to reconcile with the protein interacting with the extracellular part of SERK3.

Re: Thank you for your comments. In the revised manuscript, we also added new experiment to demonstrate that PIPeL1, PIPeL1-like, and LcPIP1 were secreted into apoplast.

First, we supplemented the western blot analysis of apoplastic fluid from *N. benthamiana* leaves agroinfiltrated with PIPeL1, PIPeL1-like, and LcPIP1. PIPeL1, PIPeL1-like, and LcPIP1 could be successfully detected in apoplastic fluid using immunological analysis (new Fig. 1b and new Fig. 4f), indicating that three proteins were indeed secreted into the apoplastic space.

Second, we supplemented the experiment to validate the secretory function of the signal peptide of PIPeL1 and PIPeL1-like by yeast signal trap assay (new Fig. 1a), and results supported that the signal peptides of PIPeL1 and PIPeL1-like possess secretory function.

Third, it has been previously reported that PsXEG1 must be targeted to the extracellular space of *N. benthamiana* tissue to induce cell death (Ma *et al.*, 2015). We noticed that expression of LcPIP1^{SP}-PsXEG1^{ΔSP} was able to induce cell death, whereas PsXEG1^{ΔSP} or LcPIP1^{SP}-RFP could not induce cell death in *N. benthamiana* (new Supplementary Fig. 11c). In addition, both LcPIP1^{SP}-PsXEG1^{ΔSP} and PsXEG1 could be detected as two isoforms (new Supplementary Fig. 11d). These results further confirmed that the signal peptide of LcPIP1 has the secretory function.

Fourth, SERK3 (BAK1) possesses both an extracellular domain (ECD) leucine-rich repeat (LRR) and an intracellular cytoplasmic domain (CD) (Jaillais *et al.*, 2011) (new Fig. 7d). In the revised manuscript, Co-IP assays showed that the extracellular domain of LcSERK3 interacted with LcPIP1, but not the cytoplasmic domain (new Fig. 7e). This result further confirms that LcPIP1 interacted with the extracellular part of LcSERK3.

References:

- Ma, Z. *et al.* A *Phytophthora sojae* glycoside hydrolase 12 protein is a major virulence factor during soybean infection and is recognized as a PAMP. *The Plant Cell*. 27, 2057-2072 (2015).
- Jaillais, Y. *et al.* Extracellular leucine-rich repeats as a platform for receptor/coreceptor complex formation. *Proceedings of the National Academy of Sciences*. 108, 8503-8507 (2011).

In figure 3d it is also noteworthy that the protein carrying the PR1-SP appears smaller than LcPIP, when it should be at least identical (if not bigger, based on the cartoon in Fig 3c). This could be explained if the PR1-SP was processed, but then the size should be the same as LcPIP1^{ΔSP}, which is not. All this needs to be clarified.

Re: First of all, we apologize, as there might be unknown issues with the *Agrobacterium tumefaciens* strains carrying C-terminal HA-tagged PR1^{SP}-LcPIP1^{ΔSP} constructs. In response to this concern, we conducted a repeated *Agrobacterium transformation* for the three plasmids (pBin-LcPIP-HA, pBin-PR1^{SP}-LcPIP1^{ΔSP}-HA, and pBin-LcPIP1^{ΔSP}-HA) and subsequently infiltrated the resulting single colonies into *N. benthamiana*. Total proteins were then extracted from *N. benthamiana* leaves at 36 hpa. Our findings demonstrated that the protein sizes of LcPIP1-HA and PR1^{SP}-LcPIP1^{ΔSP}-HA were essentially similar, and both were larger than LcPIP1^{ΔSP}-HA (new Supplementary Fig. 11b, or Author Response Figure 2). Consistent with the previous results, expression of LcPIP1 or PR1^{SP}-LcPIP1^{ΔSP} induced cell death at 3 dpa, while LcPIP1^{ΔSP} could not induce cell death in *N. benthamiana* (new Fig. 4g). These experiments were repeated three times, and we consistently obtained similar results. To address this issue, we have rectified the problem in the blots and provided the blot photos in the Source data file for further inspection.

LcPIP1^{ΔSP} appeared smaller than LcPIP1 or PR1^{SP}-LcPIP1^{ΔSP} in the manuscript. Actually, similar phenomena have been observed in other apoplastic proteins, such as expansin-like proteins (PcEXLX1), PGIP-INactivating Effector 1 (SsPINE1), aldose

1-epimerase (AEP1), and PcINF1 (Pi *et al.*, 2022; Wei *et al.*, 2022; Xu *et al.*, 2021; Liu *et al.*, 2015).

We speculate that LcPIP1 may undergo unknown post-translational modifications, resulting in an increase in the molecular weight of the protein. For example, both LcPIP1^{SP}-PsXEG1^{ΔSP} and PsXEG1 could be detected two isoforms, but not PsXEG1^{ΔSP} (new Supplementary Fig. 11d). Furthermore, we also noticed that the accumulation of the LcPIP1^{ΔSP} protein was lower. In *N. benthamiana* expressing either LcPIP1 or LcPIP1^{ΔSP}, the transcription levels of *LcPIP1* were comparable (Author Response Figure 3a). Therefore, the discrepancy of protein accumulation may be attributed to the lack of proper localization instructions in LcPIP1^{ΔSP}, which could potentially have affected its stability and lead to its degradation or instability within the cell.

Considering the diminished accumulation of LcPIP1^{ΔSP} protein relative to LcPIP1, we investigated whether LcPIP1 could still induce cell death at lower protein levels. Our findings demonstrated that LcPIP1 retained the ability to induce cell death in *N. benthamiana*, even when its protein level was reduced (Author Response Figure 3b,c).

Author Response Figure 2 (For review only) Immunoblot analysis of LcPIP1, PR1^{SP}-LcPIP1^{ΔSP} and LcPIP1^{ΔSP} expressed in *N. benthamiana* leaves.

Total proteins were extracted from *N. benthamiana* leaves at 36 hpa. Red asterisks indicated protein bands of the correct size. Ponceau S staining of Rubisco was used to indicate loading quantity of protein samples.

Author Response Figure 3 (For review only) LcPIP1 could induce cell death in *N. benthamiana* at low protein levels.

(a) qRT-PCR analyzed the transcription levels of the C-terminal region of *LcPIP1* in leaves expressing LcPIP1 or LcPIP1^{ΔSP}. Statistical analysis was conducted using the Student's *t*-test method. (b) Total proteins were extracted from *N. benthamiana* leaves at 36 hpa. Red asterisks indicated protein bands of the correct size. Ponceau S staining of Rubisco was used to indicate loading quantity of protein samples. (c) *N. benthamiana* leaves were infiltrated with *Agrobacterium tumefaciens* strains carrying HA-tagged LcPIP1 or LcPIP1^{ΔSP} constructs. Photographs were taken at 4 dpa.

References:

- Pi, L. *et al.* A G-type lectin receptor-like kinase regulates the perception of oomycete apoplastic expansin-like proteins. *J. Integr. Plant Biol.* **64**, 183-201 (2022).
- Wei, W. *et al.* A fungal extracellular effector inactivates plant polygalacturonase-inhibiting protein. *Nat. Commun.* 13, (2022).
- Xu, Y. *et al.* *Phytophthora sojae* apoplastic effector AEP1 mediates sugar uptake by mutarotation of extracellular aldose and is recognized as a MAMP. *Plant Physiol.* 187, 321-335 (2021).
- Liu, Z. *et al.* SRC2-1 is required in PcINF1-induced pepper immunity by acting as an interacting partner of PcINF1. *J. Exp. Bot.* 66, 3683-3698 (2015).

Related to this, in several CO-IP experiments, the IP for the target protein shows several bands, some of them present at the same intensity as the target protein, which are not present in controls (Figures 7 and Suppl 7). What could these proteins be and how do the authors rule out that the interaction does not take place with one of these proteins instead of with the target protein?

Re: Thanks for the comments. Previous reports have indicated that that SERK3 (BAK1) undergoes Ca²⁺-dependent proteolytic cleavage *in planta*, which plays an essential role in BAK1-regulated plant immunity (Zhou *et al.*, 2019).

In response to your concern, we have confirmed the interaction using RFP-tagged PIP1s and HA-tagged LcSERK3, NbSERK3A, NbSERK3B, and AtBAK1 (new Fig. 7c). Combined with previous experiments, our results proved that LcPIP1 interact with LcSERK3/NbSERK3a in plant (new Fig. 7).

Regarding Suppl 7 (new Supplementary Fig. 6b), we confirmed the interaction between RFP-tagged LcPIP1 and HA-tagged PIPeL1/PIPeL1-like (Fig. 2b). Furthermore, GST pull-down have further verified that the target protein, rather than degraded protein, interacted with LcPIP1 (new Supplementary Fig. 6c).

References:

- Zhou, J. *et al.* Proteolytic Processing of SERK3/BAK1 Regulates Plant Immunity, Development, and Cell Death. *Plant Physiol.* 180, 543-558 (2019).

Finally, in most cases the authors present their results as histograms with the means ±

SE of independent biological replicates. I suggest adding the data points to the histograms in order to have a better view of the dispersion of the data points, and identifying the different repeats (alternatively use boxplots or other presentation that helps visualizing dispersion). Numbers of datapoints should be indicated for all experiments. I assume that the authors confirmed that their dataset fulfils the requirements for the statistical test used.

Re: Thank you for your suggestions. We have added the data points to all the histograms.

Other suggestions

Abbreviate differentially *Phytophthora* and *Peronophthora*

Re: Thanks for pointing this out. We have abbreviated them differentially as *Phytophthora* (*Ph.*) and *Peronophthora* (*P.*)

Overall, since the authors have not demonstrated pectate lyase activity, it might be more appropriate using the terms putative or candidate pectate lyases.

Re: Thank you for your comments. We conducted analysis on the pectate lyase activity of PIPEL1, PIPEL1-like and mutants (new Fig. 1e). And LcPIP1 did not suppress the enzymatic activity of PIPEL1 or PIPEL1-like (new Supplementary Fig. 8c). The detailed experiments associated with pectate lyase activity have been added to the revised manuscript.

Overall, the authors use lesion diameter to account for pathogen infection, but lesions are not always circular, in particular for experiments on litchi. Using lesion area would be more accurate.

Re: Thanks for the suggestion. In revised manuscript, we conducted lesion area measurements using the ImageJ software for cases where lesions were not circular, ensuring a more precise assessment of pathogen infection. However, for circular lesions, we still used diameter measurements for statistical analysis. This is consistent with established method for lesion size quantification in *Phytophthora* infection studies, as documented in relevant literature (Li *et al.*, 2023; Qiu *et al.*, 2023; Yang *et al.*, 2023; Yang *et al.*, 2022).

References:

Li, F., *et al.*, Potato protein tyrosine phosphatase StPTP1a is activated by StMKK1 to negatively regulate plant immunity. *Plant Biotechnology Journal*, 2023. 21(3): p. 646-661.

- Qiu, X., *et al.*, The *Phytophthora sojae* nuclear effector PsAvh110 targets a host transcriptional complex to modulate plant immunity. *The Plant Cell*, 2023. 35(1): p. 574-597.
- Yang, K., *et al.*, The *Pythium periplocum* elicitor PpEli2 confers broad-spectrum disease resistance by triggering a novel receptor-dependent immune pathway in plants. *Horticulture Research*, 2023. 10(2).
- Yang, Y., *et al.*, A mitochondrial RNA processing protein mediates plant immunity to a broad spectrum of pathogens by modulating the mitochondrial oxidative burst. *The Plant Cell*, 2022. 34(6): p. 2343-2363.

Overall, the figure legends are limited. This is particularly true for the legends of the Supplementary Figures.

Re: Thank you for your suggestions. We have revised the manuscript to include more detailed figure legends for clearer expression.

P7, l 118-122. The claim that up-regulation of either PIPeL1 and PIPeL1-like upon knocking out the other supports functional redundancy should be played down. In Suppl Fig 1 the authors show that both genes are separated by approx. 1100 bp. This proximity in the promoters for both genes, even if they are in opposite strands, could lead to a competition for expression in the WT, and thus to an increase of the expression of one of the genes when the other is knocked down.

Re: Thanks for pointing this out. We have made adjustments to the manuscript. Firstly, we have repositioned the figure that was previously mentioned as "Fig. 1c" as a supplementary figure (now identified as Supplementary Fig. 4). Additionally, as per your suggestion, we have revised the manuscript by incorporating the relevant content into the discussion section (Lines 382-389).

P7, l 130-131 and Suppl Fig 5. In order to highlight the conservation of the Asp residues, it could be more interesting to present the alignment of PIPeLs with other PeLs from other species. Related to this, the sentence in l 130 needs to be developed.

Re: Thank you for your suggestions. We have added the alignment of homologous proteins of PIPeL1 and PIPeL1-like in 15 oomycete species (no homologous protein of PIPeL1/PIPeL1-like was identified in fungi or plants). The findings from this alignment reveal that the seven D residues are also highly conserved in 24 homologous proteins of PIPeL1 and PIPeL1-like (new Supplementary Fig. 5b). We have incorporated these relevant results into the revised manuscript (Lines 123-126).

Define positive and negative controls in Figure 2a

Re: Thanks! BD-53+AD-T and BD-Lam+AD-T were employed as the positive and negative controls, respectively. Detailed explanations of these controls can be found in the revised Methods (Lines 615-616) and the figure legend (Fig. 2a).

P11, l 208. The conserved motif VDMASG could be extended to VDMASGK/RYLW
Re: Thank you for your suggestions. Previously, we contemplated extending the conserved motif to include VDMASGK/RYLW. However, our experiments with the deletion mutant M1 (1-75 aa) showed that it induced cell death in *N. benthamiana* to a similar extent as the full-length protein. This finding suggested that K/RYLW may not be the critical domain responsible for inducing plant resistance. While the conserved role of VDMASGK/RYLW motif in other functions remains unexplored, we have decided to use the VDMASG motif as it appears to be more suitable for our current study.

P11, l 209. It is not clear to me what is the contribution to the manuscript of the phylogenetic tree presented in Figure 5a.

Re: Thanks for the comments. We have expanded the scope of the phylogenetic tree from an initial 12 plant species to now encompass 24 species, including additional members from the Sapindaceae family. The phylogenetic tree and the sequence alignment together illustrate the sequence and evolutionary relationships of LcPIP1 and its homologous proteins. Importantly, we found that 11 homologous proteins of LcPIP1, spanning different groups, showed cell death-inducing activity (new Fig.4e), suggesting that conserved role of PIP1s in inducing cell death in *N. benthamiana*. We have revised the related sentence for clearer expression (Lines 237-243).

P11, l 209. Provide accession numbers for all proteins in Suppl Figure 9.

Re: Thank you for your suggestion. We have added the accession numbers for each protein (new Supplementary table 3). These sequences were retrieved from NCBI (<https://www.ncbi.nlm.nih.gov>), Sol Genomics Network (<http://solgenomics.net/>), Sapinaceae Genomic (<http://www.sapindaceae.com/>), or UniProt (<https://www.uniprot.org/>). The detailed information is seen under "Accession numbers" in the revised Methods.

P15, l 297. NbPIP1 is upregulated upon INF-1 mediated cell death (Figure 8d) as well as upon PIPeL1 expression (Suppl Figure 11). INF1 induces defense responses and PIPeLs suppresses defense responses. How does it fit with the model of NbPIP1 function?

Re: Thank you for your question. Firstly, we need to clarify that in our study, we did not find evidence of PIPeL1 or PIPeL1-like suppressing plant defense signaling. In this study, we observed that their expression in *N. benthamiana* enhanced susceptibility to *Ph. capsici*, consistent with their role as virulence factors in

Peronophythora litchii. The pectate lyase enzyme activity of PIPeL1 and PIPeL1-like likely facilitates pathogen invasion by degrading plant cell walls, and the PIPeL1/PIPeL1-like enzymatic activity mutants lost their virulence (new Figure 1e-g). These results are consistent with previous reports of pathogen-secreted pectate lyases (Hugouvieux-Cotte-Pattat et al., 2014; Cnossen-Fassoni et al., 2013; Cho et al., 2015).

In our experiments, INF1-mediated cell death was remarkably delayed in *NbPIP1*-silenced *N. benthamiana* plants (Figure 8a). Furthermore, as a positive immune regulatory factor, LcPIP1 triggered immune activation in *N. benthamiana*. In our recently conducted experiments, we discovered that co-expressing PIPeL1^{M1}/PIPeL1-like^{M1} with LcPIP1 resulted in increased electrolyte leakage in the injection area compared to the control, suggesting that PIPeL1^{M1}/PIPeL1-like^{M1} could enhance LcPIP1-induced plant cell death (new Figure 3j).

Therefore, we believe that the response of PIP1 to both INF1 and PIPeL1/PIPeL1-like is an integral part of the plant immune response, aimed at countering pathogen invasion and activating defense pathways to restrict pathogen proliferation.

However, we have chosen to remove this result from our analysis. This decision was based on the observation of a relatively low increase induced by PIPeL1 and PIPeL1-like on *NbPIP1* (1.2 and 1.39-fold, respectively), which, while statistically significant, may not provide substantial insights or contribute significantly to our overall findings.

References:

- Hugouvieux-Cotte-Pattat, N. *et al.* Bacterial pectate lyases, structural and functional diversity. *Environ. Microbiol. Rep.* 6, 427-440 (2014).
- Cnossen-Fassoni, A. *et al.* The pectate lyase encoded by the *pecCII* gene is an important determinant for the aggressiveness of *Colletotrichum lindemuthianum*. *J. Microbiol.* 51, 461-470 (2013).
- Cho, Y. *et al.* A pectate lyase-coding gene abundantly expressed during early stages of infection is required for full virulence in *Alternaria brassicicola*. *Plos One.* 10, e127140 (2015).

As a non-native English speaker, I understand how frustrating this can be, but the manuscript would benefit from proof-reading by a native English speaker.

Re: Thanks for the suggestion. Based on your comment, a native English speaker who is a professor of plant pathology has looked at the manuscript for language issues, which is also mentioned in the Acknowledgements section.

Reviewer #2 (Remarks to the Author):

This MS describes the identification and characterisation of two secreted proteins (PIPeL1 and PIPeL1-like proteins) from *Peronophythora litchii* and their target.

In short, the authors show that:

- 1- deletion of both genes in *P. litchii* results in a loss of virulence
- 2- Said PI proteins interact with a small secreted protein (Pip1) in plants
- 3- LcPIP mediates cell death (orchestrated by a small and highly conserved motif)
- 4- Plant cell death resembles a PTI-type response that requires SERK3 and activate expression of known PTI responsive genes

Based on these observations, the authors conclude that LcPIP is a cell death inducing peptide that interacts with two PI effectors.

Overall, the work is presented nicely and the MS is of good written quality. Major comments are listed below:

Re: We thank the reviewer for the positive comments on our data and constructive suggestions, and time for reviewing our manuscript.

- It is not clear to me what the main message of the work is and how it is important. LcPIP induces cell death when over-expressed in plants and its silencing compromises immunity to virulent pathogens. This is an interesting observation (bolstered by interesting interaction data). Does this reduced immunity phenotype extend to non-pathogenic microbes? Reduction of PTI gene expression hints at a more general impairment, but this is not explored. That's where I think exploration would lead to exciting discoveries.

Re: Thank you for your comments and suggestions.

1) Our research sheds light on a novel and pivotal aspect of plant-pathogen interactions by elucidating the interaction between the pectate lyase of the plant pathogen and the novel positive immune regulator, LcPIP1.

This interaction operates within a BAK1/SERK3-dependent framework. Then, we found that loss of PIP could significantly impair plant immunity in *Arabidopsis*, and *N. benthamiana*. Our result suggested that PIP is critical and general in plant immune system. Further study might reveal PIP1 as a new layer/key regulate of complexity in the plant immune response, particularly in the PTI pathways.

In revised manuscript, we have added new results to support the function of PIP1 (new Fig. 3g-i, new Fig 4e, and new Fig 6) and tried our best to clearly describe our conclusion (Lines 476-485).

2) In our experiments, we conducted inoculation trials using a range of non-pathogenic microbes, including *P. litchii*, *Pyricularia angulate*, *Colletotrichum gloeosporioides*, and *Lasiodiplodia theobromae* on *AtPIP1* knockout *Arabidopsis* plants and *NbPIP1*-silenced *N. benthamiana* plants. However, the results showed that these non-pathogenic microbes failed to infect either *Arabidopsis* or *N. benthamiana*. Based on these materials, we have not yet found that PIP1-mediated immunity is related to non-pathogenic microbes.

Nevertheless, your insight has inspired us to begin to screen a broader range of non-pathogenic microbes or mutants to assess the potential susceptibility of *PIP1*

mutants. But it will take a long time to reveal the underlying mechanism, and we hope to reveal those findings in a future publication.

- The authors show that two PI effectors bind Litchi (and Nb) PIP AND that disabling both effectors affect virulence. The significance of these observation is not clear. What is the consequence of PIP1 binding? Disabling of PIP1 function or PAMP perception? The authors will know that knocking out two effector coding genes could cause a virulence phenotype that has little to do with LcPIP1 signalling. Infection assays that feature knockout strains and PIP1 mutants would help address that.

Re: Thanks for the comments and suggestions.

1) As reviewer mentioned, PIPEL1/PIPEL1-like possess pectate lyase activity, allowing them to degrade the plant cell wall and enhance virulence during pathogen infection. Regarding the proposal for infection assays involving knockout strains and *PIP1* mutants, we understand the significance of this approach. However, its execution poses several challenges: firstly, *P. litchii* does not infect *N. benthamiana* or Arabidopsis; second, homologous proteins of PIPEL1 and PIPEL1-like are absent in *Ph. capsici*; third, current genetic transformation methods for *Ph. infestans* and litchi are difficult, which hinders the creation of knockout strains and mutants. These challenges collectively hinder our ability to perform the suggested experiments.

2) We have conducted additional experiments to further investigate the relationship between PIPEL1/PIPEL1-like and LcPIP1. The results suggested that *LcPIP1* could be indirectly regulated by PIPEL1/PIPEL1-like in litchi (new Fig. 4b); PIPEL1^{M1}/PIPEL1-like^{M1} could enhance LcPIP1-induced cell death (new Fig. 3j); and PIPEL1/PIPEL1-like did not disrupt the interaction between LcPIP1 and LcSERK3 (new Fig. 7f,g). In addition, silencing of *NbPIP1* resulted in enhanced susceptibility of plants induced by PIPEL1 (new Fig. 3k,l). These results support the hypothesis that the interaction between PIPEL1/PIPEL1-like and PIP1s can enhance plant immunity, which is mediated by PIP1s in *N. benthamiana*.

It is possible that PIP1-mediated immunity is actively suppressed during infection and if so, understanding the mechanisms by which PI disrupts its signalling would be of great interest. As it stands, however, there is not enough data to support a credible model that informs on the action of PIP1 and (possibly) PI virulence factors. The proposed model reflect these knowledge gaps as it is fairly vague and throws up many questions.

Re: Thanks! We acknowledge the possibility that potential effectors or virulence mechanisms could actively suppress PIP1-mediated immunity during infection. In addition, it's notable that PIPEL1/PIPEL1-like also possess pectate lyase activity, allowing them to degrade the plant cell wall and enhance virulence during pathogen infection. This parallels the well-established importance of numerous cell wall-degrading enzymes in facilitating pathogen invasion (Dora *et al.*, 2022; Kubicek *et al.*, 2014). It is important to emphasize that the intricate interactions between plants and

pathogens are subject to a multitude of influences. Consequently, the mere expression of LcPIP1/NbPIP1 may not be sufficient to fully counteract the virulence potential of PIPeL1/PIPeL1-like in the natural condition.

We have new data to support virulence function of PIPeL1-mediated enzymatic activity. It is interesting how PIP1-mediated immunity is suppressed by pathogen; however, we did not find that PIPeL1/PIPeL1-like could suppress the function of PIP1.

References:

Dora, S. *et al.* Plant–microbe interactions in the apoplast: Communication at the plant cell wall. *The Plant Cell*. 34, 1532-1550 (2022).

Kubicek, C. P. *et al.* Plant cell wall-degrading enzymes and their secretion in plant-pathogenic fungi. *Annu. Rev. Phytopathol.* 52, 427-451 (2014).

Reviewer #3 (Remarks to the Author):

This manuscript describes the involvement of pectate lyases (PIPeL1 and PIPeL1-like) from *Peronophthora litchii* in pathogenicity and a novel plant cell death-inducing protein, LcPIP1, from litchi. First, the authors created single or double deletion mutants of PIPeL genes and demonstrated that the double deletion mutant impaired the pathogenicity of *P. litchi*. Next, the authors investigated the interaction of PIPeLs and PIP1s. They demonstrated that the signal peptides and the VDMASG motif of LcPIP1 were essential to induce PCD in *Nicotiana benthamiana*. Moreover, they showed that the plant immune response by LcPIP/NbPIP was triggered via SERK3. The data presented in this manuscript is very interesting and the methods used are reasonable. The manuscript, however, has several critical issues. I would recommend this manuscript for publication in *Nature Communications*, provided the authors reasonably address the following points.

Re: We thank the reviewer for the positive comments on our data and constructive suggestions, and time for reviewing our manuscript.

Line 91:

Is the identity (99.78%) ranged from 1 to 460 amino acids?

Re: We apologize for the mistake and thank the reviewer for bringing it to our attention. The "identity 99.78%" reported in the initial manuscript was based on the preliminary comparison result obtained from the NCBI blast (<https://blast.ncbi.nlm.nih.gov/Blast.cgi>). However, upon careful reevaluation based on full length, we have now corrected the identity to "94%" (Line 86).

These two pel genes can be named as PIPeL1 and PIPeL2 because the C-terminal regions of the two proteins are quite different. That makes easier to distinguish them in the manuscript. The authors don't have to change the names but I strongly

recommend.

Re: Thank you for your suggestions. However, through Illumina sequencing, we identified 18 PeL encoding genes in *P. litchii* genome, based on bioinformatic analysis (Ye *et al.*, 2016). Eighteen PeL genes were named based on the genome published by Ye *et al.*, 2016. Subsequently, in 2020, our laboratory reassembled the *P. litchii* genome using both second and third-generation sequencing technologies. During this process, we identified a protein with a sequence and transcription pattern highly similar to PIPeL1. Therefore, we designated it as PIPeL1-like.

References:

Ye, W. *et al.* Sequencing of the Litchi Downy Blight Pathogen Reveals It Is a *Phytophthora* Species with Downy Mildew-Like Characteristics. *Molecular Plant-Microbe Interactions*. 29, 573-583 (2016).

Line 125:

Was the full length of PIPeL1-like protein expressed including the unique C-terminus?

Re: Thanks for the question. Yes, we expressed the full length of PIPeL1-like protein in *N. benthamiana* leaves using agroinfiltration, and western blot (Fig. 1b or Supp. Fig. 6a) confirmed that the PIPeL1-like protein expressed including the unique C-terminus and was larger than PIPeL1, with a difference of approximately 7 KDa.

Line 142:

Which means that the increase of the susceptibility was responsible for only the degradation of pectic substances (pectate)?

Re: Thank you for your question. According to previous reports, it's well-established that most plant pathogens secrete a variety of cell wall-degrading enzymes (CWDEs) to break down plant cell walls and obtain nutrients for their own growth and infection (Kikot *et al.*, 2009). In this study, we also found that the enzymatic activity of PIPeL1/PIPeL1-like were also responsible for their virulence function (new Fig. 1e-g). However, we did not state that the susceptibility was only induced by the degradation of plant cell wall.

References:

Kikot, G. E. *et al.* Contribution of cell wall degrading enzymes to pathogenesis of *Fusarium graminearum*: a review. *Journal of Basic Microbiology*. 49, 231-241 (2009).

Do PIPeL1M2 and -likeM2 contain signal peptides?

Re: We have made revisions in the manuscript, changing the names of the D residues →A residues mutants from ' PIPeL1^{M2} and PIPeL1-like^{M2} ' to PIPeL1^{M1}, PIPeL1-like

^{M1}, respectively. The decision to make this change was driven by our observation of lower protein expression levels in mutants lacking the signal peptide (SP), which raised concerns about potential loss-of-function effects caused by reduced protein expression.

Regarding your question about signal peptides, both PIPeL1^{M1} and PIPeL1-like^{M1}, as described in the revised manuscript, do contain signal peptides. We have provided this additional explanation in Line 128.

Line 166:

Have you conducted the expression assay of LcPIP1 using litchi inoculated with a single or double deletion mutants of Pel genes?

Apparently, Pels up-regulate NbPIP1 in *N. benthamiana* from the results of Supplementary Fig. 11. Whether the expression of LcPIP1 is also regulated by Pels in litchi is intriguing.

Re: Thanks for the suggestions. Recently, we investigated the expression of *LcPIP1* in single or double deletion mutants-infected litchi (new Fig. 4b). The transcription level of *LcPIP1* displayed less upregulation in litchi infected with double knockouts at 6 and 12 hpi, as opposed to the transcription level observed upon inoculation with the wild-type strain.

However, we noticed that *LcPIP1* was up-regulated during pathogen infection. So the transcription levels of *LcPIP1* decreased during infection by double-deletion mutants, which may be caused by the weakened pathogenicity of the deletion mutants. Therefore, we have added "may be" to the conclusion that *LcPIP1* is regulated by PIPeL1 and PIPeL1-like in litchi, "These results indicate that *LcPIP1* expression might be indirectly regulated by PIPeL1 and PIPeL1-like in litchi." (Lines 209-211). This modification allows us to include or not exclude the possibility that other factors, such as the influence of pathogenicity on *LcPIP1* expression, might contribute to the observed results.

Line 233:

Single (PIP) and co-expressing (PIP/PeL) leaves showed the same lesion size, which raises the following question: PIPeL1 and -like have no effect on the infection of *P. cspici*?

I just wonder how the complex of LcPIP1/PIPeL1 is involved in pathogenicity and plant resistance. I understand that LcPIP1 indeed induced PCD in *N. benthamiana* from your data. Do you expect that PIPeLs from *P. litchii* have function to inhibit LcPIP1 from binding to SERK3 through interacting with each other in addition to degradation of pectic substances? or LcPIP is secreted to inhibit the enzymatic activity of PIPeLs in addition to the induction of immune systems? I understand that they have not completely proven yet in your work, but I believe it is necessary to describe your prediction at least. From the title of your paper, readers will imagine the effect given by the interaction between Pel and PIP toward pathogenicity and the plant immune systems.

Re: Thank you for your comments and suggestions. Our study revealed that the expression of PIPEL1/PIPEL1-like promote the infection of *Ph. capsici* and the knockout of these two genes led to reduced virulence in *P. litchii*. These results indicated that PIPEL1 and PIPEL1-like play critical roles in the virulence of *P. litchii*. However, when PIPEL1 or PIPEL1-like was co-expressed with LcPIP1 in *N. benthamiana*, LcPIP1-mediated plant defense compromised the virulence function of PIPEL1/PIPEL1-like. Silencing of *NbPIP1* resulted in enhanced susceptibility of *N. benthamiana* induced by PIPEL1 (new Fig. 3k,l), and PIPEL1^{M1}/PIPEL1-like^{M1} could enhance LcPIP1-induced cell death (new Fig. 3j).

In other word, our results unveiled the dual roles of PIPEL1/PIPEL1-like in plant-pathogen interactions: enhancing pathogen virulence through pectate lyase activity while also being targeted by LcPIP1, enhancing plant immunity. It is well known that some cell wall-degrading enzymes (CWDEs) secreted by pathogens can act as both virulence factors and inducers of plant immunity. For example, GH12 protein PsXEG1, xylanase BcXyl1, pectate lyase VdPEL1 and FsPL are known to be important virulence factors while also activating the plant PTI response (Ma *et al.*, 2017; Yang *et al.*, 2018; Wang *et al.*, 2023; Yang *et al.*, 2018).

Furthermore, according to several suggestions of the reviewer, our new results showed that LcPIP1 did not interfere with PIPEL1 or PIPEL1-like enzyme activity (new Supplementary Fig. 8c). In addition, PIPEL1/PIPEL1-like did not disrupt LcPIP1 from binding to LcSERK3 through interacting with each other (new Fig. 7f,g). However, the impact of PIPEL1/PIPEL1-like on the interaction between LcPIP1 and potential receptor-like kinases (RLKs) remains unknown, and further investigation is needed in this regard.

Regarding the title, to our knowledge, this is the first report on the interaction between pectate lyases of a plant pathogen and a plant positive immune regulator. Thus, we highlight the novel interaction between pectate lyases and the positive immune regulator PIP1.

References:

- Ma, Z. *et al.* A paralogous decoy protects *Phytophthora sojae* apoplastic effector PsXEG1 from a host inhibitor. *Science*. 355, 710-714 (2017).
- Yang, Y. *et al.* The *Botrytis cinerea* Xylanase BcXyl1 Modulates Plant Immunity. *Front. Microbiol.* 9, (2018).
- Wang, C. *et al.* Pectate Lyase from *Fusarium sacchari* Induces Plant Immune Responses and Contributes to Virulence. *Microbiol. Spectr.* 11, (2023).
- Yang, Y. *et al.* A *Verticillium dahliae* pectate lyase induces plant immune responses and contributes to virulence. *Front. Plant Sci.* 9, (2018).

Line 251:

Do you mean that the susceptibility induced by PIPEL1 is different from that by PIPEL1-like in mechanisms? If NbPIP1 has higher ability to attenuate the susceptibility induced by PIPEL1, I just wonder why leaves expressing PIPEL1 or

PIPeL1-like, expressing PIP1 as well, showed the same lesion size (Figure 1d-g and Figure 6ab).

Re: Thanks for the question.

1) In our study, when PIPeL1 or PIPeL1-like was co-expressed with LcPIP1, the *N. benthamiana* leaves showed the same lesion size (new Fig. 3a,b). However, when PIPeL1 was co-expressed with LcPIP1, the relative *Ph. capsici* biomass was significantly smaller compared to co-expressing PIPeL1-like and LcPIP1, as determined by a Student's *t*-test (new Fig. 3c). Quantification of pathogen biomass provides a more detailed assessment of *Ph. capsici* growth and proliferation during infection, suggesting that LcPIP1 might exert a more pronounced impact on attenuating the virulence function of PIPeL1 compared to PIPeL1-like.

2) We adjusted the OD₆₀₀ of the injected Agrobacterium to 0.05, 0.005, and 0.0005. Surprisingly, under lower OD₆₀₀ conditions (OD₆₀₀ = 0.005), lesions on leaves expressing PIPeL1 were significantly smaller than those on leaves expressing RFP (Author Response Figure 4a). Despite this, PIPeL1-like still promoted pathogen invasion more effectively than RFP. Furthermore, in OD₆₀₀ = 0.0005 conditions, lesions on leaves expressing PIPeL1 and PIPeL1-like were significantly smaller than those on leaves expressing RFP (Author Response Figure 4b). At low concentrations of both PIPeL1 and PIPeL1-like, they may play a major role in inducing plant resistance. This suggests that PIPeL1-like can exert its virulence effect at lower protein levels, possibly making the plant resistance response induced by NbPIP1 less susceptible to interference by PIPeL1-like.

3) *PIPeL1* showed a greater than 20-fold increase in relative expression compared to *PIPeL1-like* at 24 hpi (new Supplementary Fig. 1e), suggesting that the plant may require stronger suppression of PIPeL1 virulence. Based on above findings, we hypothesize that LcPIP1 may more strongly inhibit the virulence function of PIPeL1. This could explain why silencing *NbPIP1* did not significantly increase the virulence of PIPeL1-like.

4) Additionally, in *NbPIP1*-silenced plants, specific plant resistance-related genes may be upregulated and specifically recognize PIPeL1-like. However, these underlying mechanisms remain enigmatic and warrant further investigation. In the revised manuscript, we added additional discussion regarding this concern (Lines 421-429).

Author Response Figure 4 (For review only) The resistance of *N. benthamiana* to *Ph. capsici* is influenced by varying concentrations of Agrobacterium carrying PIPeL1 or PIPeL1-like expressing plasmids.

(a) Disease induced by different concentration of *A. tumefaciens* ($OD_{600} = 0.05, 0.005,$ or 0.0005). *N. benthamiana* leaves expressing PIPeL1, PIPeL1-like, or RFP were inoculated with *P. capsici*. Lesion development was measured at 2 days post-inoculation (dpi). (b) Confirmation of protein accumulation. Total proteins were extracted from *N. benthamiana* leaves at 36 hpa. Red asterisks indicated protein bands of the correct size. Ponceau S staining of Rubisco was used to indicate loading quantity of protein samples.

If you know, please let me know. Is the homolog of INF1 from *P. capsici* secreted in inoculated *N. benthamiana*?

Re: INF1 is a canonical ELI-1 elicitor secreted by *Ph. infestans*, and it could induce cell death, enhancing plants' resistance to *Phytophthora* infection. Elicitors are a family of highly conserved extracellular proteins in oomycete-specific (Qutob *et al.*, 2003; Liu *et al.*, 2015; Jiang *et al.*, 2006). Additionally, *Ph. capsici* also secretes the homolog of INF1 (Liu *et al.*, 2015), known as PcINF1, which triggers cell death in

pepper and *N. benthamiana* (Wang *et al.*, 2018). We have added this information in revised manuscript (Lines 296-298).

References:

- Qutob, D. *et al.* Variation in structure and activity among elicitors from *Phytophthora sojae*. *Mol. Plant Pathol.* 4, 119-124 (2003).
- Liu, Z. *et al.* SRC2-1 is required in PcINF1-induced pepper immunity by acting as an interacting partner of PcINF1. *J. Exp. Bot.* 66, 3683-3698 (2015).
- Jiang, R. H. Y. *et al.* Ancient Origin of Elicitor Gene Clusters in *Phytophthora* Genomes. *Mol. Biol. Evol.* 23, 338-351 (2006).
- Wang, Z. *et al.* Osa-miR164a targets OsNAC60 and negatively regulates rice immunity against the blast fungus *Magnaporthe oryzae*. *The Plant Journal.* 95, 584-597 (2018).

Line 295:

What do you think is the reason why the expression of WRKY8 was not influenced? Please state about it in Discussion.

Re: Thank you for your question. Notably, WRKY7, WRKY8, WRKY9, and WRKY11 have been identified as crucial components in INF1-mediated PTI ROS bursts (Adachi *et al.*, 2015). Moreover, these WRKY proteins, including WRKY8, are known to redundantly contribute to the transactivation of RBOHB upon INF1 treatment (Adachi *et al.*, 2015; Adachi *et al.*, 2016). In the *NbPIP1*-silenced plants, the observed delays in INF1-induced cell death, rather than complete inhibition. These findings could potentially explain the absence of transcriptional differences in WRKY8 between *NbPIP1* and *GUS*-silenced plants. Further investigations are warranted to unravel the precise regulatory mechanisms governing *WRKY8* expression and its potential interaction with *NbPIP1* in the context of the immune response. In the revised manuscript, we added additional discussion regarding this concern (Lines 455-465).

References:

- Adachi, H. *et al.* WRKY Transcription Factors Phosphorylated by MAPK Regulate a Plant Immune NADPH Oxidase in *Nicotiana benthamiana*. *The Plant Cell.* 27, 2645-2663 (2015).
- Adachi, H., *et al.* *Nicotiana benthamiana* MAPK-WRKY pathway confers resistance to a necrotrophic pathogen *Botrytis cinerea*. *Plant Signal. Behav.* 11, e1183085 (2016).

Line 331:

Please add 'in *N. benthamiana*' at the end of the sentence.

Re: We have revised the sentence.

Line 365:

As you mentioned, NbPIP1 may interact with INF1 to trigger the immune systems. Again, how about the interaction (binding) between PeL1 and PIP1? What does this interaction cause? Just to avoid the degradation of cell walls?

Re: Thanks for the questions. In our experiments, INF1-mediated cell death was remarkably delayed in *NbPIP1*-silenced *N. benthamiana* plants (Fig. 8a). Furthermore, as a positive immune regulatory factor, LcPIP1 triggers cell death accompanied by reactive oxygen species (ROS) accumulation, callose deposition, and induction of defense genes in *N. benthamiana*. LcPIP1 interacted and perceived with PIPeL1 and PIPeL1-like to increase host resistance (new Fig. 3). Therefore, our results showed that the response of PIP1 to both INF1 and PIPeL1/PIPeL1-like will activate plant defense signaling to restrict pathogen proliferation.

As mentioned above, we conducted additional experiments. The results indicate that LcPIP1 did not affect the pectate lyase enzyme activity of PIPeL1 and PIPeL1-like. Therefore, currently, there is no evidence to support the inhibition of PIPeL1/PIPeL1-like-mediated cell wall degradation by LcPIP1.

Line 379:

You mentioned that NbPIP1 attenuated the susceptibility induced by PIPeL1 in line 251 (Figure 6g,h). But here, you describe that NbPIP1-silencing did not affect the susceptibility quoting Figure 6g,h. It is confusing.

Re: Thank you for your comments and questions.

1) In our revised manuscript, we conducted experiments involving the knockout of the homologous gene *AtPIP1* in Arabidopsis. These *atpip1* knockout mutants showed an increased susceptibility to *Ph. capsici* infection (new Fig. 3g-i).

2) The silencing of gene using VIGS results in a reduction in its expression, rather than complete elimination. As shown in Figure 3d, the expression level of *NbPIP1* remained at 15.5% of the control (TRV-*GUS*). Consequently, the plant retained a certain level of recognition and defense capabilities, albeit at a diminished level.

3) Furthermore, it's crucial to consider that PIPeL1 and PIPeL1-like lack a homologous protein in *Ph. capsici* (new Supplementary Fig. 5b). PIPeL1 directly interacted with NbPIP1 (Fig. 2b,c), possibly enhancing the virulence function of PIPeL1 in *NbPIP1* silencing plants.

4) In light of the complex network of plant defense responses, it is conceivable that the effects of *NbPIP1* silencing may be partially countered by compensatory mechanisms involving other proteins within the intricate defense signaling network. This intricate web of interactions and compensatory responses could collectively contribute to the observed absence of a significant difference in pathogen resistance between the *NbPIP1*-silenced plants and the control.

5) In the revised manuscript, we added additional discussion regarding this concern (Lines 414-421).

Line 386:

Can you say PIPeLs trigger the immune systems?

If so, leaves expressing PIPeLs should show resistance against *P. capsici* somehow.

Your sentence from lines 386 to 388 sounds like it is happening in litchi.

I suggest that you describe the roles of PeLs from the pathogen end and those of PIP1 from the plant end, and on top of that, what the interaction of PeLs and PIP1 brings about.

Please distinguish clearly whether you are describing about *N. benthamiana* or litchi.

Re: Thanks for the questions and suggestions.

1) Our results demonstrate that PIPeL1^{M1}/PIPeL1-like^{M1} could enhance LcPIP1-induced cell death (new Fig. 3j). Furthermore, *LcPIP1* expression may be modulated by PIPeL1/PIPeL1-like in litchi. These results support that the interaction between PIPeL1/PIPeL1-like and PIP1s can enhance plant immunity, which is mediated by PIP1s. However, it's important to note that PIPeL1/PIPeL1-like also possess pectate lyase activity (new Fig. 1e), allowing them to degrade the plant cell walls and enhance virulence during pathogen infection. This parallels the well-established importance of numerous cell wall-degrading enzymes in facilitating pathogen invasion (Dora *et al.*, 2022; Kubicek *et al.*, 2014).

Variations in the expression levels of PIPeL1 and PIPeL1-like lead to diverse impacts on plant resistance against *Ph. capsici* (Author Response Figure 4). Specifically, when PIPeL1/PIPeL1-like accumulation was at a lower level, plant resistance took precedence, while higher accumulation shifted the balance toward a dominance of virulence.

It is important to emphasize that the intricate interactions between plants and pathogens are subject to a multitude of influences. Consequently, the mere expression of LcPIP1/NbPIP1 may not consistently suffice to fully counteract the virulence potential of PIPeL1/PIPeL1-like under natural conditions. Pathogens can employ a diverse array of strategies, including the secretion of other effectors, to subvert or manipulate host resistance mechanisms.

This study presents a novel perspective on plant-pathogen interactions, specifically highlighting the interaction between novel plant's immune regulator, PIP1 and pathogen-secreted PeLs (PIPeL1/PIPeL1-like). We acknowledge that the mechanistic intricacies behind this interaction warrant further exploration. In ongoing research, we are using Co-IP and LC-MS/MS analyses to identify downstream resistance-associated proteins directly engaged by the interaction between PIPeL1/PIPeL1-like and LcPIP1. Particular attention is being given to proteins potentially associated with salicylic acid (SA), aiming to shed light on the potential relationships between LcPIP1 and systemic acquired resistance (SAR).

2) According to the reviewer's suggestion, we have made efforts to provide a more explicit conclusion regarding the role of the PIPeL1/PIPeL1-like-PIP1 module in the plant-pathogen interaction (Lines 476-485).

3) We have made a concerted effort to maintain a clear distinction between the descriptions of *N. benthamiana* and litchi throughout the entirety of our manuscript.

References:

Dora, S. *et al.* Plant-microbe interactions in the apoplast: Communication at the plant cell wall. *The Plant Cell*. 34, 1532-1550 (2022).

Kubicek, C. P. *et al.* Plant cell wall-degrading enzymes and their secretion in plant-pathogenic fungi. *Annu. Rev. Phytopathol.* 52, 427-451 (2014).

Reviewer #1 (Remarks to the Author):

The authors made a great job addressing the different points I raised upon reviewing a first version of the manuscript. The revised manuscript includes new results that strengthen the claims made by the authors and the manuscript reads much better, in part thanks to the reorganization of the presentation of the results. I will be happy to recommend the paper for publication. I couldn't help noticing some minor points, mostly formal, that I believe should be corrected in order to improve the quality of the manuscript. I don't need to see the revised version.

In figures 3C and 4A, the authors provide two different statistical analysis of the results (One-way ANOVA and Student-test). This leads to some cases where two samples are not significantly different with one test but different with the other (see for example Fig 4A, 0h and 1.5h), because each test addresses different questions. This makes the presentation quite confusing. I believe that for the sake of clarity the results of just one test should be presented, which, as I see it, should be the ANOVA for Fig 3C and the Student's test for Fig 4A.

In Figure 6, based on Figure 6C the INF1 used in the experiments is RFP-tagged. Would it be possible to clarify this in Fig 6B and in the Figure legend?

In Figure 8A, the presence of INF1 is detected with an anti-RFP antibody, but in the previous version of the manuscript the same western-blot reported INF1 detected with an anti-HA antibody. I suppose there was a mistake in the first version of the manuscript?

Related to this, a Supplementary Table with all constructions used in the manuscript and a one-line description of their use will be very helpful.

The writing of the manuscript is greatly improved, but I couldn't help noticing two sentences in the figure legends that need to be corrected:

L 1094: PIPeL1 or PIPeL1-like could not disrupted LcPIP1-LcSERK3 interaction

L 1120: qRT-PCR analysis relative expression of NbPIP1 gene

Following the change in analysis of the qPCR results, figure 8E shows that NbPIP1 expression is also induced by Agroinfiltration at 36 hpi. I believe this is interesting because it supports the idea of NbPIP1 being induced upon biotic stress. Would it be possible to add this to the discussion?

Also, would it be possible to discuss the fact that INF1-mediated responses are affected by PIP1, but not FLS2-mediated responses, considering that both are SERK3-dependent?

Reviewer #3 (Remarks to the Author):

In the revised manuscript, I recognized that most of the issues pointed out by reviewers have been improved.

I, to some extent, understood that the interaction of PIPels and PIP1 in *N. benthamiana* is necessary for the stronger immune response from the results of the new Fig. 3c, j, l., although even solo PIP1 can induce a certain degree of immune response.

The authors mentioned in their response letter that 'we hypothesized that LcPIP1 may more strongly inhibit the virulence function of PIPeL1'. In the assay of enzyme activity (Sup. Fig.8c), however, LcPIP1 did not affect the enzymatic activity of PIPeL1 and -like. What does 'the virulence function of PIPeL1' mean in the above sentence? In line 478, the authors also state that 'The attenuation of PIPeL1/PIPeL1-like virulence by LcPIP1/NbPIP1 may result from immune activation following their interaction'. If PIP1 doesn't affect the enzyme activity of Pels, what else does it affect?

In Fig. 8f, it would be easier to understand, if an unknown receptor for the complex of PIPeL1/LcPIP1 is also shown.

Response to Reviewers

REVIEWER COMMENTS

Reviewer #1 (Remarks to the Author):

The authors made a great job addressing the different points I raised upon reviewing a first version of the manuscript. The revised manuscript includes new results that strengthen the claims made by the authors and the manuscript reads much better, in part thanks to the reorganization of the presentation of the results. I will be happy to recommend the paper for publication. I couldn't help noticing some minor points, mostly formal, that I believe should be corrected in order to improve the quality of the manuscript. I don't need to see the revised version.

Re: Thank you very much for recommending acceptance of our revised manuscript and the helpful comments on the manuscript.

In figures 3C and 4A, the authors provide two different statistical analysis of the results (One-way ANOVA and Student-test). This leads to some cases where two samples are not significantly different with one test but different with the other (see for example Fig 4A, 0h and 1.5h), because each test addresses different questions. This makes the presentation quite confusing. I believe that for the sake of clarity the results of just one test should be presented, which, as I see it, should be the ANOVA for Fig 3C and the Student's test for Fig 4A.

Re: Thanks for your suggestions. We applied a single statistical method to analyze the data in Fig 3c, Fig 4a, and Supplementary Fig. 1c. Specifically, we conducted a one-way ANOVA for Fig 3c and employed the Student's t-test for both Fig 4a and Supplementary Fig. 1c.

In Figure 6, based on Figure 6C the INF1 used in the experiments is RFP-tagged. Would it be possible to clarify this in Fig 6B and in the Figure legend?

Re: Thanks. We have revised Fig 6b for clarity and have also revised the Figure legend to describe that the INF1 is tagged with RFP. Similar modifications have been made to Fig 8a,c,e and the Figure legend.

In Figure 8A, the presence of INF1 is detected with an anti-RFP antibody, but in the previous version of the manuscript the same western-blot reported INF1 detected with an anti-HA antibody. I suppose there was a mistake in the first version of the manuscript?

R: Yes, there was a mistake in the first version of the manuscript. We have checked it

again for correct description.

Related to this, a Supplementary Table with all constructions used in the manuscript and a one-line description of their use will be very helpful.

Re: Thank you for your suggestion. We have added a Supplementary Table (new Supplementary Table 5) that provides a list of all constructions used in this manuscript, along with a brief one-line description of their use.

The writing of the manuscript is greatly improved, but I couldn't help noticing two sentences in the figure legends that need to be corrected:

L 1094: PIPeL1 or PIPeL1-like could not disrupted LcPIP1-LcSERK3 interaction

L 1120: qRT-PCR analysis relative expression of NbPIP1 gene

Re: Thanks for pointing this out. We have made the corrections to the two sentences. In Line 1094, we have revised the sentence to "PIPeL1/PIPeL1-like did not disrupt the LcPIP1-LcSERK3 interaction" (new Lines 1103-1104). In Line 1120, it has been revised to "qRT-PCR analyzed the expression levels of *NbPIP1* gene in *N. benthamiana* leaves expressing INF1 or RFP at 12, 24, and 36 hpa" (new Lines 1130-1132).

Following the change in analysis of the qPCR results, figure 8E shows that NbPIP1 expression is also induced by Agroinfiltration at 36 hpi. I believe this is interesting because it supports the idea of NbPIP1 being induced upon biotic stress. Would it be possible to add this to the discussion?

Re: Thanks for your suggestions. We have revised the manuscript by incorporating the relevant content into the discussion section, "In this study, we observed a significant up-regulation of *LcPIP1*, *AtPIP1*, and *NbPIP1* when plants were exposed to biotic stress conditions, including infections by *P. litchii*, *Ph. capsici*, or *A. tumefaciens* (Figure 4a, Figure 8e, and Supplementary Fig. 7c). These results provide evidence for the involvement of PIP1s in plant responses to biotic challenges" (Line 412-416).

Also, would it be possible to discuss the fact that INF1-mediated responses are affected by PIP1, but not FLS2-mediated responses, considering that both are SERK3-dependent?

Re: Thank you for your question. The differential impact of PIP1 on INF1-mediated and FLS2-mediated responses implies pathway-specific regulatory roles. Although both pathways involve SERK3, distinct downstream signaling components or regulatory mechanisms may account for the differential responses mediated by INF1 and FLS2. This distinction merits further examination and discussion to reveal the underlying molecular processes.

Reviewer #3 (Remarks to the Author):

In the revised manuscript, I recognized that most of the issues pointed out by reviewers have been improved.

I, to some extent, understood that the interaction of PIPeLs and PIP1 in *N. benthamiana* is necessary for the stronger immune response from the results of the new Fig. 3c, j, l., although even solo PIP1 can induce a certain degree of immune response.

The authors mentioned in their response letter that ‘we hypothesized that LcPIP1 may more strongly inhibit the virulence function of PIPeL1’. In the assay of enzyme activity (Sup. Fig.8c), however, LcPIP1 did not affect the enzymatic activity of PIPeL1 and -like. What does ‘the virulence function of PIPeL1’ mean in the above sentence? In line 478, the authors also state that ‘The attenuation of PIPeL1/PIPeL1-like virulence by LcPIP1/NbPIP1 may result from immune activation following their interaction’. If PIP1 doesn't affect the enzyme activity of Pels, what else does it affect?

Re: Thank you for your question. We realize that the statement about LcPIP1 and PIPeL1/PIPeL1-like in our previous response and manuscript was inaccurate. In this revision, we corrected it as "we hypothesize that LcPIP1 may more strongly inhibit PIPeL1-mediated *N. benthamiana*'s susceptibility, which potentially explains why silencing *NbPIP1* did not significantly increase the virulence of PIPeL1-like".

Furthermore, we have conducted a thorough review of our manuscript to ensure a precise description and discussion of the relationship between PIP1 and susceptibility mediated by PIPeL1/PIPeL1-like. Our revised statement is as follows: "LcPIP1/NbPIP1's role in attenuating PIPeL1/PIPeL1-like induced *N. benthamiana*'s susceptibility to *Ph. capsici* may be resulted from immune activation following their interaction. Specifically, PIP1s induce plant defense, and the interaction of PIPeL1/PIPeL1-like and PIP1s in *N. benthamiana* may induce a stronger immune response".

In Fig. 8f, it would be easier to understand, if an unknown receptor for the complex of PIPeL1/LcPIP1 is also shown.

Re: Thanks for your suggestions. We have included an unknown receptor for the complex of PIPeL1/LcPIP1 and PIPeL1-like/LcPIP1 in the revised Figure 8f.